# BARYCENTRIC ALIGNMENT OF MUTUALLY DISENTANGLED MODALITIES

## ABSTRACT

Discovering explanatory factors of user preferences behind behavioral data has gained increasing attention. As collected behavioral data is often highly sparse, mining other data modalities, e.g., texts, for interest factors and then correlating them with those from behavioral data could provide a pathway to improve recommendation. Nonetheless, two challenges prevail. For one, the unordered set nature of discovered factors and the unavailability of prior alignment information causes a challenge to align revealed interest factors from two modalities. For another, it demands a tailored method to effectively transfer knowledge between interest factors from mutually related modalities. To resolve this, we regard discovered interest factors from ratings and texts as supporting points of two discrete measures. Then, their alignment is formulated as an optimal transport problem, finding an optimal mapping between two probability masses. Next, the mapping probability serves not only as the prior information but also as input of barycentric strategy to match and fuse interest factors, effectively tranferring user preferences between mutually disentangled modalities. Experiments on real-world datasets verify the advantage of the proposed method over a series of baselines.

## 1 INTRODUCTION

Variational Autoencoder (VAE)-based disentangled representation learning has demonstrated a strong performance on recommendation task. Current studies ranged from dimension-wise disentanglement Higgins et al. (2017); Burgess et al. (2018); Liang et al. (2018) to both intention and dimension disentanglement Ma et al. (2019b) of user preferences. Recently, as gaining user trusts into recommender systems has become more important, side information has been incorporated into VAE-based disentangled models Guo et al. (2022); Wang et al. (2023a;b) to enhance interpretability of user preferences besides improving recommendation accuracy.

Discovering interest factors from side information of multiple modalities then incorporating them into recommendation models is a direction of emergent pertinence. One the one hand, preference signals behind side information could complement those mined from user consumption behaviors, resulting in more expressive interest representations. On the other hand, uncovered interest factors from side information naturally act as 'interpreter' of preferences as people are able to interpret meaning behind textual or visual content. Thus, mapping interest factors from behavioral data with those discovered from side information is a feasible way to gain insights into user preferences.

Therefore, we study the problem of disentangling and aligning interest factors across modalities, particularly for content-aware recommendation involving text and rating modalities. Nonetheless, simultaneously discovering and mapping interest factors from both rating and text modalities is fundamentally different from disentangling preferences from a single modality and therefore poses two challenges. For one, as interest factors from both modalities are discovered unsupervisedly, while their alignment information is not available in advance, it requires a proper method to derive interest factor alignment effectively in unsupervised setting. For another, the unordered set nature of discovered preference factors makes it challenging to transfer user interest information between factors as well as fuse their representations to obtain more expressive ones.

To tackle the challenges, we introduce a novel model BANDVAE, standing for Barycentric Alignment of Mutually Disentangled Modalities with Variational AutoEncoder. BANDVAE differentiates itself with three primary aspects. *Firstly*, we jointly discover multiple user preference

factors behind ratings and texts by resorting to prototype-based representation learning. *Secondly*, by regarding discovered factors from modalities as supporting points of two discrete measures, we cast their alignment problem into an optimal transport (OT) problem. OT enables finding alignment probabilities of interest factors unsupervisedly via solving an entropic regularized optimal transport problem using Sinkhorn algorithm, which effectively addresses the first challenge. *Thirdly*, we formulate two approaches to resolve knowledge (user preferences) transfer problem. For one, an alignment probability-guided regularization term is included in the learning objective. For another, a barycentric mapping strategy maps rating (text) factors onto text (rating) space then incorporating the mapped factors into decoder to reconstruct rating (text) input, enabling mutual transference of supervision signals from two modalities. *In addition*, our model is capable of dealing with users demonstrating different behaviors in each modality, i.e., the numbers of interest factors of modalities differ.

**Contributions.** Our key contributions are three-fold. *First*, we study the problem of discovering and matching disentangled factors across modalities to uncover the explanatory factors behind users' adoptions. *Second*, we introduce a novel model BANDVAE, aligning interest factors from ratings and texts via optimal transport and transferring knowledge between uncovered factors via alignment probability-guided regularization and barycentric mapping. *Third*, we conduct extensive experiments quantitatively and qualitatively on real-world datasets to verify the advantages of BANDVAE.

## 2 RELATED WORK

**VAE-based disentangled representation learning.** Uncovering hidden explanatory factors behind data results in robust representations and enables modeling complex data Bengio et al. (2013). Variational AutoEncoder or VAE is a popular method offering representation disentanglement. Early works including Higgins et al. (2017); Burgess et al. (2018); Kim & Mnih (2018); Chen et al. (2018); Locatello et al. (2019) aim to achieve dimension-level disentanglement. Later, Ma et al. (2019b) disentangles user preferences in recommender systems in both dimension and intention levels. Follow-up works extend MacridVAE by incorporating side information to improve recommendation accuracy and interpretability Guo et al. (2022); Wang et al. (2023a). Our work follows this line of research yet distinguishes itself by innovatively incorporating *optimal transport* for aligning disentangled rating and text factors. Wang et al. (2023b) disentangles user preferences from both consumption and social environments, which is fundamentally different from ours.

**Textual content-aware recommendation.** Early works Wang & Blei (2011); Wang et al. (2015); Kim et al. (2016); Li & She (2017) explore various methods to model item textual content then regularize it with item representation. Ma et al. (2019a) leverages attention to model text and gated mechanism to fuse textual content into autoencoder based recommendation model. Recently, pre-trained language models (PLM), e.g., Devlin et al. (2019) have been explored to generate text representation as input to recommendation model Zhang et al. (2021a); Zhou & Shen (2023); Zhou et al. (2023). Regarding VAE-based recommendation, Zhang et al. (2020) disentangles preferences from collaborative and content features. Follow-up works incorporate textual content into VAE both in non-disentangled Zhu & Chen (2022) and disentangled fashions Tran & Lauw (2022); Guo et al. (2022). We follow disentangled direction to discover multiple factors of user preferences from ratings and texts then align those factors to enhance recommendation and interpretability. What sets our work apart is the optimal transport-based method for aligning and fusing interest factors from modalities. Compressing texts into single vector using PLM might lose the granularity of texts. Thus it may be unable to interpret user preferences via text, which is out of our interest in this paper.

**Optimal transport and its applications.** Optimal Transport (OT) provides an elegant way to measure two probability distributions as well as transport a point from a distribution to another Peyré & Cuturi (2019). Sinkhorn algorithm is a widely adopted method Cuturi (2013); Genevay et al. (2018); Peyré & Cuturi (2019) to compute optimal transport plan as it offers a differentiable and GPU-friendly solution, which enables various applications. Courty et al. (2014; 2017) apply OT for domain adaptation. Singh & Jaggi (2020) fuses different models' layers. Zhang et al. (2021b) aligns multiple query and key matrices in multi-head attention. Sander et al. (2022) improves attention matrix in Transformer. Cao et al. (2022) improves knowledge graph modeling by fusing multimodal data via OT. Wu et al. (2023) designs OT-inspired regularization term to improve topic modeling. Zhang et al. (2023) explores OT for object-centric learning. Our novelty is characterized by aligning and fusing mutually disentangled user interests from ratings and texts for recommendation.

# 3 PRELIMINARIES AND NOTATIONS

Let $\mathcal{U}$ be the set of $M$ users indexed by $u$, and $\mathcal{V}$ be the set of $N$ items indexed by $i$. The interactions between users and items are stored in $\mathbf{R} \in \{0,1\}^{M \times N}$. For user $u$, let $\mathbf{y}^u \in \{0,1\}^N$ be her historical interactions with items. $\mathbf{y}^u$ is the $u^{th}$ row of $\mathbf{R}$. $\mathbf{y}^u_i = 1$ indicates an observed interaction between user $u$ and item $i$, otherwise $\mathbf{y}^u_i = 0$. For item $i$, let $\mathbf{w}^i \in \mathbb{R}^W$ be the tf-idf representation of its textual content. $W$ is the number of words in the vocabulary. Let $\mathbf{t}^u \in \mathbb{R}^W$ be textual vector associated with user $u$, obtained by averaging tf-idf vectors of user $u$'s adopted items, i.e., $\mathbf{t}^u = \frac{\sum_i \mathbf{y}^u_i \mathbf{w}^i}{\sum_i \mathbf{y}^u_i}$. Let $\mathbf{H} \in \mathbb{R}^{N \times d}$ be the embedding matrix of $N$ items, which is the weight of decoder of rating channel in Figure 1. The encoder of rating channel maintains a context matrix $\mathbf{C}^y \in \mathbb{R}^{N \times D}$ used to derive the user representation from $\mathbf{y}^u$. For text channel in Figure 1, the weight of decoder is denoted by $\mathbf{E}^{W \times d}$, which stores $W$ $d$-dimensional vectors of $W$ words in the vocabulary. The encoder of text channel also includes a context matrix $\mathbf{C}^t \in \mathbb{R}^{W \times D}$, which is used to derive user representation from $\mathbf{t}^u$.

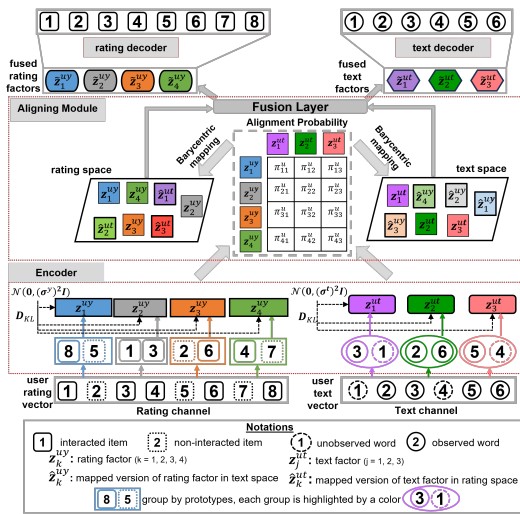

Figure 1: Illustration of our model BANDVAE. Dashed squares and circles are not included in deriving user interests.

Our goal is to reveal user preferences underlying $\mathbf{y}^u$ and $\mathbf{t}^u$. To achieve this, we seek factorized user representations from $\mathbf{y}^u$ and $\mathbf{t}^u$, denoted as $\mathbf{z}^{uy}$ and $\mathbf{z}^{ut}$, respectively. Concretely, $\mathbf{z}^{uy} = \{\mathbf{z}^{uy}_k\}^K_{k=1}$ assuming $K$ user interest factors underlying $\mathbf{y}^u$. Similarly, $\mathbf{z}^{ut} = \{\mathbf{z}^{ut}_j\}^J_{j=1}$ consists of $J$ interest factors behind $\mathbf{t}^u$. Next, we align these rating and text factors via optimal transport. The target is two-fold. For one, aligning and fusing interest factors increases their expressiveness thanks to combining knowledge mined from two modalities. For another, mapping rating factors onto text space improves interpretability as textual content is human-understandable.

# 4 METHODOLOGY

In this section, we present the proposed model BANDVAE illustrated in Figure 1. BANDVAE includes three main components: **a) prototype-based encoder** discovers $K$ rating factors and $J$ text factors of user interests; **b) aligning module** includes three sub tasks: i) estimates the alignment probabilities between rating and text factors; ii) maps rating (text) factors onto text (rating) space; iii) fuses rating (text) factors with their mapped version; **c) decoder** receives fused rating and text factors from **b** to reconstruct observed user-item interactions and user associated text.

## 4.1 PROTOTYPE-BASED ENCODER

**Rating encoder.** Let $f^{enc}_{rating}(\mathbf{y}^u, \mathbf{m}^y, \mathbf{H}, \mathbf{C}^y, \tau, \sigma^y)$ be encoder of rating channel in Figure 1. The input includes user $u$'s adoptions $\mathbf{y}^u$, prototypes $\mathbf{m}^y \in \mathbb{R}^{K \times d}$, item embedding $\mathbf{H} \in \mathbb{R}^{N \times d}$, context matrix $\mathbf{C}^y \in \mathbb{R}^{N \times D}$, temperature $\tau$, and hyper-parameter $\sigma^y$. First, we group $N$ items into $K$ clusters by prototypes, producing an item assignment score matrix $\mathbf{A}^y \in \mathbb{R}^{N \times K}$.

$$\mathbf{A}^y = \eta\left(\frac{\mathbf{H} \cdot (\mathbf{m}^y)^T}{\tau \cdot ||\mathbf{H}||_2 \cdot ||\mathbf{m}^y||_2}, axis = `K`\right) \quad (1)$$

$axis = `K`$ means the operator is performed along axis of $K$ prototypes. Following Ma et al. (2019b), $\eta$ is Gumbel-Softmax Jang et al. (2017); Maddison et al. (2017), i.e., if item $i$ belongs to cluster $k$, then $\mathbf{A}^y_{ik}$ is close to 1 and $\mathbf{A}^y_{ik'}, \forall k' \neq k$, is near zero. $\mathbf{A}^y_{ik}$ is based on cosine similarity between item $i$ embedding $\mathbf{H}_i$ and prototype $\mathbf{m}^y_k$ to prevent mode collapse, i.e., items are mainly

associated with single prototype Ma et al. (2019b). Small $\tau$ is to obtain skewed distribution of assignment score of $\mathbf{A}_i^y$ of item $i$. Next, we aggregate user adopted items belonging to cluster $k$ then estimate parameters $\boldsymbol{\mu}_k^{uy}$ and $\boldsymbol{\sigma}_k^{uy}$ of Gaussian distribution of user $u$'s $k^{th}$ rating factor.

$$(\mathbf{r}_k^{uy}, \mathbf{o}_k^{uy}) = g^y \left( \frac{\sum_i \mathbf{y}_i^u \mathbf{A}_{ik}^y \mathbf{C}_i^y}{\sqrt{\sum_i \mathbf{y}_i^u (\mathbf{A}_{ik}^y)^2}} \right) \Longrightarrow \boldsymbol{\mu}_k^{uy} = \frac{\mathbf{r}_k^{uy}}{||\mathbf{r}_k^{uy}||_2}; \ \ \boldsymbol{\sigma}_k^{uy} = \sigma^y \cdot exp(-\frac{1}{2}\mathbf{o}_k^{uy}) \tag{2}$$
$$\mathbf{z}_k^{uy} \sim \mathcal{N}(\boldsymbol{\mu}_k^{uy}, [diag(\boldsymbol{\sigma}_k^{uy})]^2) \ \forall k = 1, 2, ..., K$$

Here, $g^y : \mathbb{R}^D \to \mathbb{R}^{2d}$ is function parameterized by neural network. $\sigma^y$'s value is around 0.1, following Ma et al. (2019b). Rating factor $k$ representation of user $u$ is sampled from Gaussian distribution with estimated parameters. Assuming the independence between rating factors of user $u$, we have $q(\mathbf{z}^{uy}|\mathbf{y}^u, \mathbf{A}^y) = \prod_{k=1}^K \mathcal{N}(\boldsymbol{\mu}_k^{uy}, [diag(\boldsymbol{\sigma}_k^{uy})]^2)$, which is called variational distribution, approximating intractable posterior distribution $p(\mathbf{z}^{uy}|\mathbf{y}^u, \mathbf{A}^y)$. $q(\mathbf{z}^{uy}|\mathbf{y}^u, \mathbf{A}^y)$ is matched with prior distribution $p(\mathbf{z}^{uy}) = \mathcal{N}(\mathbf{0}, (\sigma^y)^2 \mathbf{I})$ via Kullback-Leibler divergence ($D_{KL}^y$). As $p(\mathbf{z}^{uy})$ is factorized, $D_{KL}^y$ imposes micro-disentanglement, i.e., disentangled between dimensions of representation sampled from $q(\mathbf{z}^{uy}|\mathbf{y}^u, \mathbf{A}^y)$. $D_{KL}^y(q(\mathbf{z}^{uy}|\mathbf{y}^u, \mathbf{A}^y)||p(\mathbf{z}^{uy}))$ will be plugged into Equation 12 for optimization.

**Text encoder.** Let $f_{text}^{enc}(\mathbf{t}^u, \mathbf{m}^t, \mathbf{E}, \mathbf{C}^t, \tau, \sigma^t)$ be the encoder of text channel in Figure 1. The input includes user $u$'s texts $\mathbf{t}^u$, prototypes $\mathbf{m}^t \in \mathbb{R}^{K \times d}$, word embedding $\mathbf{E} \in \mathbb{R}^{W \times d}$, context matrix $\mathbf{C}^t \in \mathbb{R}^{N \times D}$, temperature $\tau$, and hyper-parameter $\sigma^t$. The procedure of text encoder is similar to that of rating encoder. Due to limited space, we present the detailed derivation in supplementary materials. The output of text encoder include $J$ user interest factors $\{\mathbf{z}_j^{ut}\}_{j=1}^J$, word assignment matrix $\mathbf{A}^t$, and regularization term $D_{KL}^t(q(\mathbf{z}^{ut}|\mathbf{t}^u, \mathbf{A}^t)||p(\mathbf{z}^{ut}))$ with $p(\mathbf{z}^{ut}) = \mathcal{N}(\mathbf{0}, (\sigma^t)^2 \mathbf{I})$.

## 4.2 ALIGNING MODULE

As $\{\mathbf{z}_k^{uy}\}_{k=1}^K$, $\{\mathbf{z}_j^{ut}\}_{j=1}^J$ are derived unsupervisedly and their alignment information is unavailable, this requires us to discover a proper solution capable of handling their alignment. In addition, this also demands an interest transfer method between these factors so as to improve recommendation accuracy and user interest interpretability. Thus, we regard discovered rating and text factors of user $u$ as supporting points of two discrete measures and cast their alignment task as an optimal transport (OT) problem. For one, OT provides a distribution matching framework capable of estimating alignment probabilities $\pi^u$ between rating and text factors. For another, $\pi^u$ guides the knowledge transfer between two modalities and serves as input of barycentric mapping.

### 4.2.1 INTEREST FACTORS ALIGNMENT AS TRANSPORTATION PROBLEM

Let $\gamma^{uy} = \sum_{k=1}^K p_k^y \delta_{\mathbf{z}_k^{uy}}$, $\gamma^{ut} = \sum_{j=1}^J p_j^t \delta_{\mathbf{z}_j^{ut}}$ be discrete measures of user $u$'s rating and text factors, respectively. $\delta_{\mathbf{z}_k^{uy}}$ and $\delta_{\mathbf{z}_j^{ut}}$ are Dirac delta functions at $\mathbf{z}_k^{uy} \in \mathbb{R}^d$ and $\mathbf{z}_j^{ut} \in \mathbb{R}^d$, respectively. $p_k^y$ and $p_j^t$ are probability masses of $k^{th}$ rating factor and $j^{th}$ text factor, respectively. $p_k^y$ and $p_j^t$ belong to probability simplex, i.e., $\sum_{k=1}^K p_k^y = 1$ and $\sum_{j=1}^J p_j^t = 1$. Since we do not have access to the ground truth distribution and $\{\mathbf{z}^{uy}\}_{k=1}^K$, $\{\mathbf{z}^{uy}\}_{j=1}^J$ are derived unsupervisedly, we follow Wu et al. (2023) to set $p_k^y = 1/K$ and $p_j^t = 1/J$, which is uniformly distributed. Let $\mathcal{P}^u$ be the set of alignment probabilities between two distributions $\gamma^{uy}$ and $\gamma^{ut}$, we have

$$\mathcal{P}^u = \{\pi^u \in \mathbb{R}_+^{K \times J} | \pi^u \mathbf{1}_K = p^y, (\pi^u)^T \mathbf{1}_J = p^t\} \tag{3}$$

where $\mathbf{1}_K, \mathbf{1}_J$ are $K$ and $J$-dimensional one vectors. Following Cuturi (2013), a regularized optimal transport problem is formulated to make it tractable

$$\pi^u = \underset{\pi^u \in \mathcal{P}^u}{\arg\min} \langle \pi^u, \mathbf{S}^u \rangle_F - \epsilon \cdot Entropy(\pi^u) \tag{4}$$

The first term is the Frobenius dot product between $\pi^u$ and the cost matrix $\mathbf{S}^u \in \mathbb{R}^{K \times J}$, $\mathbf{S}_{kj}^u = ||\mathbf{z}_k^{uy} - \mathbf{z}_j^{ut}||_2^2$ and $\langle \pi^u, \mathbf{S}^u \rangle_F = \sum_{k,j} \pi_{kj}^u \mathbf{S}_{kj}^u$. The second term $Entropy(\pi^u) = \sum_{k,j} -\pi_{kj}^u log(\pi_{kj}^u)$ is the entropy of $\pi^u$, which is added to make the problem tractable. $\epsilon$ is a hyper-parameter of entropic regularization term. Small $\epsilon$ results in skewed distribution while large $\epsilon$ leads to relatively uniform distribution in $\pi^u$. The goal of Equation 4 is to find $\pi^u$ that minimizes the total transporting cost from rating factors to text factors of user $u$, where the cost is based on Euclidean distance between two factors.

To compute the optimal alignment probability $\pi^u$ in Equation 4, we resort to Sinkhorn algorithm Cuturi (2013). Generally, this algorithm works by alternatively calculating $\mathbf{u}$ and $\mathbf{v}$, which are two scaling vectors until convergence as presented in Algorithm 1. Then, the alignment probability between rating factors $\{\mathbf{z}_k^{uy}\}_{k=1}^K$ and text factors $\{\mathbf{z}_j^{ut}\}_{j=1}^J$ of user $u$ is

$$\pi^u = diag(\mathbf{u})exp(-\mathbf{S}^u/\epsilon)diag(\mathbf{v}) \quad (5)$$

Algorithm 1 is efficient as it is differentiable and matrix multiplication is highly supported on GPU. Besides mapping text factors to rating space to improve recommendation, $\pi^u$ enables deriving interpretability of user interests, in which rating factors are mapped to text space for interpretation.

---

**Algorithm 1:** Alignment probability derivation

**Input:** $\{\mathbf{z}_k^{uy}\}_{k=1}^K, \{\mathbf{z}_j^{uy}\}_{j=1}^J, \epsilon$

**Output:** $\pi^u$

1: $\mathbf{S}_{kj}^u = ||\mathbf{z}_k^{uy} - \mathbf{z}_j^{ut}||_2^2, \mathbf{S}^u \in \mathbb{R}^{K \times J}$
2: $\mathbf{B}^u = exp(-\mathbf{S}^u/\epsilon)$
3: Initialize $\mathbf{v} \leftarrow \mathbf{1}_J$
4: **while** not converged **do**
5: $\quad \mathbf{u} = \frac{1}{K}\frac{\mathbf{1}_K}{\mathbf{B}^u\mathbf{v}}; \mathbf{v} \leftarrow \frac{1}{J}\frac{\mathbf{1}_J}{(\mathbf{B}^u)^T\mathbf{u}}$
6: **end while**
7: **return** $\pi^u = diag(\mathbf{u})\mathbf{B}^u diag(\mathbf{v})$

---

### 4.2.2 INTEREST TRANSFER BETWEEN FACTORS OF MODALITIES

We formulate two approaches for user interest transfer between rating and text factors: i) regularization term guided by optimal alignment probabilities; ii) mapping rating (text) factors to text (rating) space via barycentric strategy so that supervision signals of modalities can be mutually transferred.

**Alignment probability guided regularization,** of which optimization is guided by $\pi^u$

$$\mathcal{L}_u^{OT} = \sum_{k=1}^K \sum_{j=1}^J \pi_{kj}^u \cdot ||\mathbf{z}_k^{uy} - \mathbf{z}_j^{ut}||_2^2 \quad (6)$$

Thanks to the probabilities captured by $\pi^u$, the optimization will focus on transferring user interests between most probably aligned factors. $\mathcal{L}_u^{OT}$ is included in 4.3 for optimization. Note that regularization-based interest transfer has been explored in Wang et al. (2015); Li & She (2017); Tran & Lauw (2022), both in non-disentangled and disentangled fashions. However, none of these is guided by alignment probabilities, which is crucial for recommendation as shown in Section 5.2.

**Mapping and fusing.** We incorporate rating (text) factors into text (rating) decoder so they could capture both modality signals, which requires mapping rating factors to text space (vice versa).

*Barycentric Mapping.* We resort to barycentric strategy Perrot et al. (2016); Courty et al. (2017) to map interest factors. Note that $\pi_{kj}^u$ tells us how much probability mass of $\mathbf{z}_k^{uy}$ to be transferred to $\mathbf{z}_j^{ut}$. Thus, we exploit this information to find the transformation of rating factors in text space by solving $\hat{\mathbf{z}}_k^{uy} = \arg\min_{\mathbf{s}^t \in \mathbb{R}^d} \sum_j \pi_{kj}^u c(\mathbf{s}^t, \mathbf{z}_j^{ut})$ . In which, $\hat{\mathbf{z}}_k^{uy}$ be the transformation of $\mathbf{z}_k^{uy}$ onto text space and $c(\cdot, \cdot)$ is $l_2$ distance cost function. Following Courty et al. (2017), the solution for $\hat{\mathbf{z}}_k^{uy}$ is

$$\hat{\mathbf{z}}_k^{uy} = diag(\pi_k^u \mathbf{1}_J)^{-1}\pi_k^u \mathbf{z}^{ut} \quad (7)$$

where $\mathbf{z}^{ut} = \{\mathbf{z}_j^{ut}\}_{j=1}^J \in \mathbb{R}^{J \times d}$. We repeat Equation 7 $\forall k = 1, 2, ..., K$ to obtain $\{\hat{\mathbf{z}}_k^{uy}\}_{k=1}^K$. Similarly, we compute $\hat{\mathbf{z}}_j^{ut}$, the transformation of text factor $\mathbf{z}_j^{ut}$ onto rating space, as follows

$$\hat{\mathbf{z}}_j^{ut} = diag((\pi^u)_j^T \mathbf{1}_K)^{-1}(\pi^u)_j^T \mathbf{z}^{uy} \quad (8)$$

where $\mathbf{z}^{uy} = \{\mathbf{z}_k^{uy}\}_{k=1}^K \in \mathbb{R}^{K \times d}$. Equation 8 is repeated $\forall j = 1, 2, ..., J$ to obtain $\{\hat{\mathbf{z}}_j^{ut}\}_{j=1}^J$.

*Adaptively Fusing.* We fuse $\{\mathbf{z}_k^{uy}\}_{k=1}^K$ with transformed versions $\{\hat{\mathbf{z}}_k^{uy}\}_{k=1}^K$ as input of rating decoder. Thus, rating signals are transferred explicitly into text space via $\{\hat{\mathbf{z}}_k^{uy}\}_{k=1}^K$. As each user's decision bases differently on modalities, we design a fusion layer with adaptive personalized weight $\rho^u$

$$\tilde{\mathbf{z}}_k^{uy} = \mathbf{z}_k^{uy} + \rho^{uy} \cdot \hat{\mathbf{z}}_k^{uy}, \quad \forall k = 1, 2, ..., K \text{ with } \rho^{uy} = log(1 + exp(\mathbf{W}^T[\mathbf{z}_k^{uy}; \hat{\mathbf{z}}_k^{uy}])) \quad (9)$$

$\rho^{uy}$ is the positive fusion weight via *softplus*, i.e., $softplus(x) = log(1 + exp(x))$. $\mathbf{W} \in \mathbb{R}^{2d \times 1}$ is a projection layer and ; is concatenation. Similarly, a fusion step is also performed for text factors.

$$\tilde{\mathbf{z}}_j^{ut} = \mathbf{z}_j^{ut} + \rho^{ut} \cdot \hat{\mathbf{z}}_j^{ut}, \quad \forall j = 1, 2, ..., J \text{ with } \rho^{ut} = log(1 + exp(\mathbf{W}^T[\mathbf{z}_j^{ut}; \hat{\mathbf{z}}_j^{ut}])) \quad (10)$$

Next, we normalize fused factors to unit-length, i.e., $\tilde{\mathbf{z}}_k^{uy} = \tilde{\mathbf{z}}_k^{uy}/||\tilde{\mathbf{z}}_k^{uy}||_2$ and $\tilde{\mathbf{z}}_j^{ut} = \tilde{\mathbf{z}}_j^{ut}/||\tilde{\mathbf{z}}_j^{ut}||_2$. Then, $\tilde{\mathbf{z}}^{uy} = \{\tilde{\mathbf{z}}_k^{uy}\}_{k=1}^K$ and $\tilde{\mathbf{z}}^{ut} = \{\tilde{\mathbf{z}}_j^{ut}\}_{j=1}^J$ go to corresponding decoder.

### 4.3 DECODER

**Rating decoder.** Let $f_{rating}^{dec}(\tilde{\mathbf{z}}^{uy}, \mathbf{A}^y, \mathbf{H}, \tau)$ be the decoder of rating channel, which accepts user $u$'s fused rating factors $\{\tilde{\mathbf{z}}_k^{uy}\}_{k=1}^K$, item assignment score matrix $\mathbf{A}^y$, item embedding matrix $\mathbf{H}$, and temperature $\tau$ as inputs. Decoder predicts the probability of interaction between user $u$ and item $i$ as

$$p(\mathbf{y}_i^u|\tilde{\mathbf{z}}^{uy}, \mathbf{A}^y) = \frac{f^r(\tilde{\mathbf{z}}^{uy}, \mathbf{H}_i)}{\sum_{i'=1}^N f^r(\tilde{\mathbf{z}}^{uy}, \mathbf{H}_{i'})} \ ; \ f^r(\tilde{\mathbf{z}}^{uy}, \mathbf{H}_i) = \sum_{k=1}^K \mathbf{A}_{ik}^y \cdot exp(\frac{\tilde{\mathbf{z}}_k^{uy} \cdot \mathbf{H}_i}{\tau \cdot ||\tilde{\mathbf{z}}_k^{uy}||_2 \cdot ||\mathbf{H}_i||_2}) \quad (11)$$

The minimizing objective for user $u$ includes cross-entropy loss to match the predicted interaction probability $p(\mathbf{y}^u)$ with observed interactions $\mathbf{y}^u$ and KL divergence term derived from Section 4.1.

$$\mathcal{L}_u^y = \sum_{i=1}^N -\mathbf{y}_i^u ln \ p(\mathbf{y}_i^u) + \beta D_{KL}^y(q(\mathbf{z}^{uy}|\mathbf{y}^u, \mathbf{A}^y)||p(\mathbf{z}^{uy})) \quad (12)$$

**Text decoder.** Let $f_{text}^{dec}(\tilde{\mathbf{z}}^{ut}, \mathbf{A}^t, \mathbf{E}, \tau)$ be the decoder of text channel. The inputs include user $u$'s fused text factors $\tilde{\mathbf{z}}^{ut}$, word assignment score matrix $\mathbf{A}^t$, word embedding matrix $\mathbf{E}$, and temperature $\tau$. The predicted probability of a word $w$ appearing in textual information of user $u$ is

$$p(\mathbf{t}_w^u|\tilde{\mathbf{z}}^{ut}, \mathbf{A}^t) = \frac{f^t(\tilde{\mathbf{z}}^{ut}, \mathbf{E}_w)}{\sum_{w'=1}^W f^t(\tilde{\mathbf{z}}^{ut}, \mathbf{E}_{w'})} \ ; \ f^t(\tilde{\mathbf{z}}^{ut}, \mathbf{E}_w) = \sum_{j=1}^J \mathbf{A}_{wj}^t \cdot exp(\frac{\tilde{\mathbf{z}}_j^{ut} \cdot \mathbf{E}_w}{\tau \cdot ||\tilde{\mathbf{z}}_j^{ut}||_2 \cdot ||\mathbf{E}_w||_2}) \quad (13)$$

Similarly, the minimizing objective of user $u$ includes cross-entropy term to match predicted probability $p(\mathbf{t}^u)$ with observed textual information $\mathbf{t}^u$ and KL divergence term derived from Section 4.1.

$$\mathcal{L}_u^t = \sum_{w=1}^W -\mathbf{t}_w^u ln \ p(\mathbf{t}_w^u) + \beta D_{KL}^t(q(\mathbf{z}^{ut}|\mathbf{t}^u, \mathbf{A}^t)||p(\mathbf{z}^{ut})) \quad (14)$$

**Final learning objective.** BANDVAE minimizes $\mathcal{L} = \frac{1}{||\mathcal{B}||}\sum_{u\in\mathcal{B}} \mathcal{L}_u^y + \lambda_t \cdot \mathcal{L}_u^t + \lambda_r \cdot \mathcal{L}_u^{OT}$. In which, $\lambda_t$ and $\lambda_r$ are hyper-parameters of $\mathcal{L}_u^t$ and $\mathcal{L}_u^{OT}$. Training procedure is included in supplements.

## 5 EXPERIMENTS

**Datasets.** We use three real-world datasets: **Citeulike-a**[1] (5,551 users; 16,980 items; 204,986 interactions; 8,000 words) contains interactions between users and scientific articles. **MovieLens**[2] (15,000 users; 7,892 items; 1,005,820 interactions; 8,000 words) includes movie ratings of user. **Amazon Cell Phones**[3] (25,500 users; 17,989 items; 285,047 interactions; 8,000 words) contains user' reviews on Cell Phones & Accessories category of Amazon dataset. We follow Ma et al. (2019b) and Zhu & Chen (2022) to collect and pre-process data. We construct training, validation and test sets by randomly selecting a subset of users for each, following strong generalization strategy in Ma et al. (2019b). Data and code are included in supplementary materials for reproducibility.

**Baselines.** We compare BANDVAE against VAE-based recommendation models, as they are capable of predicting interactions for new users not appearing in training set in strong generalization setting; including those only consider behavioral data (**MacridVAE** Ma et al. (2019b), **RecVAE** Shenbin et al. (2020)) and those involving text (**MDCVAE** Zhu & Chen (2022), **TopicVAE** Guo et al. (2022), **ADDVAE** Tran & Lauw (2022) and **SEM-MacridVAE** Wang et al. (2023a)). Among these, **RecVAE** and **MDCVAE** are non-disentangled while **MacridVAE, TopicVAE, ADDVAE, SEM-MacridVAE** are disentangled. Descriptions of these baselines can be found in the supplementary.

**Hyper-parameter settings.** We use grid search to choose hyper-parameters for baselines based on performance on validation set. Finally, we re-train these baselines with chosen ones and report performance on test set. For BANDVAE, hyper-parameter setting is presented in the supplementary.

**Recommendation evaluation metrics.** We report Recall and Normalized Discounted Cumulative Gain (NDCG) at top 10 and 50 with full-ranking strategy, i.e., test item is ranked against all items in the space. Harmonic mean of Recall and NDCG is also included to measure the overall performance.

---

[1] http://wanghao.in/CDL.htm

[2] https://grouplens.org/datasets/movielens/

[3] https://nijianmo.github.io/amazon/index.html

Table 1: Recommendation performance comparison. $\Delta_{hm}$ is harmonic mean of reported metrics. The highest results are boldfaced while the runners-up are underlined. Unit of number is percentage (%). $\star$ denotes statistical significance of the boldfaced w.r.t. the underlined. On MovieLens, where models' performance vary, BANDVAE's numbers are statistically significant on some metrics, e.g., the statistically significant gap between BANDVAE and ADDVAE on R@10, N@10 and N@50.

| Model | Citeulike-a | | | | | Cell Phones | | | | | MovieLens | | | | |
|---|---|---|---|---|---|---|---|---|---|---|---|---|---|---|---|
| | R@10 | R@50 | N@10 | N@50 | $\Delta_{hm}$ | R@10 | R@50 | N@10 | N@50 | $\Delta_{hm}$ | R@10 | R@50 | N@10 | N@50 | $\Delta_{hm}$ |
| MDCVAE | 22.43 | 38.72 | 20.97 | 26.45 | 25.66 | 4.34 | 9.60 | 3.38 | 4.87 | 4.79 | 14.03 | 29.75 | 11.98 | 18.13 | 16.43 |
| TopicVAE | 17.00 | 37.78 | 17.54 | 23.84 | 21.71 | 5.31 | 11.59 | 4.23 | 6.00 | 5.90 | 14.27 | 31.90 | 13.01 | 19.54 | 17.43 |
| RecVAE | 21.46 | 38.39 | 22.43 | 27.27 | 25.99 | 3.77 | 8.79 | 2.92 | 4.34 | 4.20 | **14.45** | 32.78 | 13.02 | 19.80 | 17.62 |
| MacridVAE | 21.92 | 43.00 | 22.95 | 29.21 | 27.27 | 5.82 | 11.96 | 4.84 | 6.58 | 6.51 | 14.25 | 32.28 | 12.74 | 19.49 | 17.32 |
| SEM-MacridVAE | 22.25 | 42.52 | 23.42 | 29.41 | 27.56 | 5.13 | 10.91 | 4.23 | 5.88 | 5.77 | 14.17 | 31.59 | **13.36** | 19.77 | 17.57 |
| ADDVAE | 23.44 | 43.89 | 24.12 | 30.23 | 28.57 | 5.76 | 11.96 | 4.90 | 6.65 | 6.54 | 14.01 | **32.95** | 12.62 | 19.63 | 17.25 |
| BANDVAE | **23.80**$^\star$ | **44.70**$^\star$ | **24.55**$^\star$ | **30.72**$^\star$ | **29.05** | **6.21**$^\star$ | **13.05**$^\star$ | **5.06**$^\star$ | **7.00**$^\star$ | **6.92** | 14.40 | 32.92 | 13.15 | **20.01**$^\star$ | **17.71** |

## 5.1 RECOMMENDATION PERFORMANCE COMPARISON

Table 1 compares the recommendation performance. Evidently, BANDVAE achieves the strongest performance on Citeulike-a and Cell Phones. On MovieLens, BANDVAE achieves better accuracy than strong baselines w.r.t. *3 out of 4* metrics and the highest harmonic mean of those metrics.

Taking a detailed look at the data, there are three data-dependent key takeaways. *First*, on Citeulike-a and Cell Phones, text-aware models, including ADDVAE, SEM-MacridVAE, TopicVAE (on Cell Phones) generally work better than using behavioral data only models MacridVAE, RecVAE, which supports the importance of textual content on recommendation accuracy. BANDVAE is better than all these baselines by a large margin, showing effectiveness of optimal transport-based alignment. *Second*, disentangled models, e.g., ADDVAE, SEM-MacridVAE, MacridVAE, have overall higher performance than non-disentangled ones, e.g., RecVAE, MDCVAE, on Citeulike-a and Cell Phones. Thus, disentangling user preferences has a crucial influence on recommendation performance. *Third*, on MovieLens, while baselines' performance vary across metrics, BANDVAE maintains the competitive performance, improving 3 out of 4 metrics by clear margins. MovieLens results show that not only disentangling user preferences is crucial but also the alignment of disentangled factors from ratings and texts. BANDVAE has a blend of these two, explaining its own effectiveness.

## 5.2 MODEL ANALYSIS

Table 2: Comparing methods producing $\pi^u$

| Method | Citeulike-a | | Cell Phones | | MovieLens | |
|---|---|---|---|---|---|---|
| | R@10 | N@10 | R@10 | N@10 | R@10 | N@10 |
| *Sinkhorn* | **23.80** | **24.55** | **6.21** | **5.06** | **14.40** | **13.15** |
| *Soft-match* | 23.31 | 24.17 | 6.18 | 5.00 | 14.15 | 12.92 |
| *Diagonal* | 22.60 | 23.39 | 5.81 | 4.78 | 13.94 | 12.88 |

Table 3: Comparing interest transfer methods

| Method | Citeulike-a | | Cell Phones | | MovieLens | |
|---|---|---|---|---|---|---|
| | R@10 | N@10 | R@10 | N@10 | R@10 | N@10 |
| *Proposed* | **23.80** | **24.55** | **6.21** | **5.06** | **14.40** | **13.15** |
| *Mapping & Fusing* | 23.31 | 24.28 | 6.10 | 4.95 | 14.19 | 12.97 |
| *Regularization* | 22.18 | 23.14 | 5.77 | 4.88 | 14.18 | 12.82 |

**Alignment probability.** $\pi^u$ lies at the heart of BANDVAE, providing probabilities to align rating and text factors of user $u$ as well as guide the interest transfer between modalities. Thus, we study other alternatives to understand the derivation of $\pi^u$. Table 2 reports the results.

- *Sinkhorn* leverages Sinkhorn algorithm as presented in Section 4.2.1

- *Soft-match* generates alignment probabilities by normalizing (negative) distance between disentangled factors from two modalities, i.e., $\pi_{kj}^u = \frac{exp(-||\mathbf{z}_k^{uy} - \mathbf{z}_j^{ut}||_2^2)/\epsilon}{\sum_{k=1}^K \sum_{j=1}^J exp(-||\mathbf{z}_k^{uy} - \mathbf{z}_j^{ut}||_2^2)/\epsilon}$.

- *Diagonal* assumes $k^{th}$ rating factor aligned with $k^{th}$ text factor, i.e., $\pi_{kj}^u = 1/K$ if $k = j$, otherwise $\pi_{kj}^u = 0$. This approach is only applicable when $K = J$.

First, *Sinkhorn* implicitly models the alignment between interest factors via alternatively scaling rows and columns of exponential negative distance matrix in Algorithm 1. It is proven that $\pi^u$ converges to optimal solution of transport problem Peyré & Cuturi (2019). Thus, *Sinkhorn* is more effective than *soft-match* normalizing exponential negative distance over all pairs of interest factors, which may result in skewed distribution, i.e., probability mass mainly concentrates on the most similar pair rather than distributing to multiple actually matched pairs of interest factors. Second, *soft-match* achieves clearly higher accuracy than *diagonal*, highlighting the importance of modeling pair-wise alignment between interest factors. Contrarily, *diagonal* makes too strong assumption about the alignment between interest factors, which hurts the accuracy.

**Interest transfer methods.** Table 3 reports the contribution interest transfer methods.

- *Proposed* includes both regularization (Section 4.2.2) and mapping and fusing (Section 4.2.2).

- *Mapping & Fusing* only includes mapping and fusing for interest transfer (no regularization).

- *Regularization* only involves regularization for interest transfer (no mapping and fusing).

There are two key takeaways. First, the two proposed interest transfer methods complement each other to boost performance. For another, *mapping & fusing* has a stronger effect than *regularization*, demonstrating that mutually transfering signals is more beneficial than regularizing interest representations. Second, the contribution of regularization method is not trivial. Combining with Table 2, we imply that it comes from estimating informative alignment probabilities via Sinkhorn algorithm.

<table>
<tr><td colspan="7">Table 4: Comparing fusion methods</td></tr>
<tr><td rowspan="2">Method</td><td colspan="2">Citeulike-a</td><td colspan="2">Cell Phones</td><td colspan="2">MovieLens</td></tr>
<tr><td>R@10</td><td>N@10</td><td>R@10</td><td>N@10</td><td>R@10</td><td>N@10</td></tr>
<tr><td>*Adaptive*</td><td>**23.80**</td><td>**24.55**</td><td>**6.21**</td><td>**5.06**</td><td>**14.40**</td><td>**13.15**</td></tr>
<tr><td>*Mean*</td><td>23.35</td><td>24.35</td><td>6.06</td><td>4.93</td><td>14.17</td><td>12.98</td></tr>
<tr><td>*Mean-T*</td><td>22.89</td><td>23.80</td><td>5.50</td><td>4.28</td><td>14.04</td><td>12.75</td></tr>
</table>

<table>
<tr><td colspan="7">Table 5: Effect of $\mathcal{L}_u^t$ on recommendation task</td></tr>
<tr><td rowspan="2">Setting</td><td colspan="2">Citeulike-a</td><td colspan="2">Cell Phones</td><td colspan="2">MovieLens</td></tr>
<tr><td>R@10</td><td>N@10</td><td>R@10</td><td>N@10</td><td>R@10</td><td>N@10</td></tr>
<tr><td>with $\mathcal{L}_u^t$</td><td>23.80</td><td>24.55</td><td>6.21</td><td>5.06</td><td>14.40</td><td>13.15</td></tr>
<tr><td>without $\mathcal{L}_u^t$</td><td>23.52</td><td>24.36</td><td>6.19</td><td>4.98</td><td>14.07</td><td>12.82</td></tr>
</table>

**Fusion method (Section 4.2.2).** Table 4 compares our fusion layer with two alternatives.

- *Adaptive* learns the adaptive fusion weight $\rho^u$ for each user $u$ as presented in Section 4.2.2.

- *Mean* takes the average of rating (text) factors and their corresponding transformed version in text (rating) space, i.e., $\tilde{\mathbf{z}}_k^{uy} = \frac{1}{2}(\mathbf{z}_k^{uy} + \hat{\mathbf{z}}_k^{ut})$ and $\tilde{\mathbf{z}}_j^{ut} = \frac{1}{2}(\mathbf{z}_j^{ut} + \hat{\mathbf{z}}_j^{uy})$.

- *Mean-T* works similarly to *mean* but operating on two *transformed versions* of factors and sharing the final factors for both channels, i.e., $\tilde{\mathbf{z}}_k^{uy} = \frac{1}{2}(\hat{\mathbf{z}}_k^{uy} + \hat{\mathbf{z}}_k^{ut}) = \tilde{\mathbf{z}}_k^{ut}$ ($k$ in place of $j$ for text factors).

First, our *adaptive* layer outperforms two fusion alternatives. Secondly, by contrasting *adaptive* and *mean*, we imply that learning a personalized fusion weight is favorable as each user makes decision differently based on modalities. Thirdly, *mean* is better than *mean-T*, indicating that sharing factor representations for modalities is limited as user preferences on these modalities might not be identical. Furthermore, residual connection between rating (text) factors with their transformed version in text (rating) domain, is also important, dropping it causes harmful effect. It is noted that while some of alignment probability derivation and fusion method have been adopted in existing works, e.g., *diagonal* and *Mean-T* in Tran & Lauw (2022), *mean* in Singh & Jaggi (2020), our proposed alternatives are better than these methods, explaining the strong performance of BANDVAE.

**Efficiency analysis.** We study the efficiency of BANDVAE by recording the training and inference time (in second). BANDVAE takes 1.7525s, 6.1622s, 3.1120s to complete a training epoch on Citeulike-a, Cell Phones and MovieLens, respectively. For contrast, we record training time of the most similar baseline ADDVAE. The training time of ADDVAE on Citeulike-a, Amazon Cell Phones and MovieLens are 1.6856s, 5.6698s and 2.9653s, respectively. Clearly, BANDVAE only requires slightly higher training time than ADDVAE yet achieving better performance by a large margin. Regarding inference time, we have the same observation, i.e., 0.0375s, 0.1230s, 0.0546s of BANDVAE compared to 0.0344s, 0.1070s, 0.0544s of ADDVAE on chosen datasets Citeulike-a, Cell Phones and MovieLens. *Overall*, the analysis supports that BANDVAE is efficient.

**Interpretability.** We study alignment probability of a user in Figure 2 and the corresponding outputs of decoder of BANDVAE in Figure 3 to qualitatively evaluate whether BANDVAE has the capability of producing interpretable user interest representations.

*First*, there is a visible staggered pattern in Figure 2. Thus, we can induce a proximate one-to-one mapping between interest factors to interpret their meaning. *Second*, rating factor 3 and 4 in Figure 2 are distinguishable, which is also demonstrated in Figure 3. Concretely, rating factor 3 are phone accessories, e.g., back holder & case and rating factor 4 seeks for composite phone case. Regarding rating factor 1 and 2, there is a degree of correlation between these two as shown in both Figure 2 and 3. Whilst rating factor 1 refers to phone wallet case and rating factor 2 talks about mobile phone and

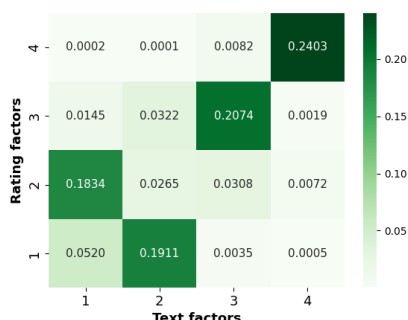

Figure 2: Alignment probability of user produced by BANDVAE on Cell Phones dataset.

camera, we observe that a wallet case is in top items of rating factor 2 and a camera shutter retrieved by rating factor 1. *Third*, outputs of text decoder has potential capability of capturing semantic meaning of rating decoder's outputs. For instance, in Figure 3, top predicted words from text factor 3 highlight the key features of items retrieved by rating factors 3, e.g., phone, case, Samsung galaxy. Similar observations can be made from other text factors. Thus, text interest factors can be potentially used to explain user preferences from rating factors. Note that interest factors from modalities are discovered unsupervisedly, which might lead to some uninterpretable interest factors, motivating us to explore other modalities to improve interpretable user interest discovery.

**Effect of $\mathcal{L}_u^t$.** We study the effect of $\mathcal{L}_u^t$ on recommendation accuracy in Table 5. Obviously, turning off effect of $\mathcal{L}_u^t$ hurts recommendation accuracy, showing its benefit. We intuit that text signals embeds user preferences into text interest factors, benefiting the alignment step.

**Effect of $\epsilon$ in Equation 4.** Experimental evidence on influence of $\epsilon$ on derivation of $\pi^u$ is shown in supplements. First, there is a trade-off between recommendation and interpretability. While recommendation favors $\epsilon > 0.2$, smaller $\epsilon$ benefiting interpretability negatively affect recommendation. Second, $\epsilon$ is data-dependent w.r.t. recommendation accuracy.

**Study the values of $K$ and $J$.** We vary number of rating factors $K$ and that of text factors $J$ and record recommendation accuracy included in supplements. Key takeaways are first, $K$ and $J$ are data-dependent; second, BANDVAE is capable of dealing with the case when the number of user interests between two modalities differ while existing work ADDVAE, is unable due to their overly strict assumption.

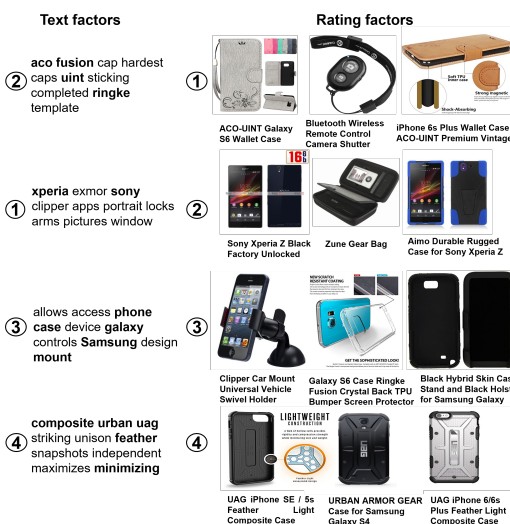

Figure 3: Visualization of predicted words and items (each has image and title) of text and rating decoders. Factor alignment is based on probability in Figure 2. Words in item titles are boldfaced.

## 6 CONCLUSION

We study the problem of aligning interest factors from mutually disentangled modalities to uncover explanatory factors behind user preferences. Then we introduce a novel model BANDVAE which aligns disentangled interest factors via optimal transport and transfer knowledge between factors via alignment probability guided regularization and barycentric mapping.

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
