# A Supplementary Materials

## A.1 Text Encoder

Let $f_{text}^{enc}(\mathbf{t}^u, \mathbf{m}^t, \mathbf{E}, \mathbf{C}^t, \tau, \sigma^t)$ be the encoder of text channel in Figure 1. The input includes user $u$'s associated text $\mathbf{t}^u$, a set of prototypes $\mathbf{m}^t \in \mathbb{R}^{K \times d}$, word embedding matrix $\mathbf{E} \in \mathbb{R}^{W \times d}$, context matrix $\mathbf{C}^t \in \mathbb{R}^{N \times D}$, temperature hyper-parameter $\tau$, and hyper-parameter $\sigma^t$.

First, $W$ words are grouped into $J$ clusters in a soft manner, producing word assignment score matrix $\mathbf{A}^t \in \mathbb{R}^{W \times J}$.

$$\mathbf{A}^t = \eta(\frac{\mathbf{E} \cdot (\mathbf{m}^t)^T}{\tau \cdot ||\mathbf{E}||_2 \cdot ||\mathbf{m}^t||_2}, axis = `J`) \tag{15}$$

$axis = `J`$ means the operator is performed along axis of $J$ prototypes. Similar to *rating encoder*, we have $\eta$ as Gumbel-Softmax trick Jang et al. (2017); Maddison et al. (2017). Next, we aggregate user associated words belonging to cluster $j$ to estimate the parameters $\boldsymbol{\mu}_k^{ut}$ and $\boldsymbol{\sigma}_k^{ut}$.

$$(\mathbf{r}_j^{ut}, \mathbf{o}_j^{ut}) = g^t(\frac{\sum_w \mathbf{t}_w^u \mathbf{A}_{wj}^t \mathbf{C}_w^t}{\sqrt{\sum_w \mathbf{t}_w^u (\mathbf{A}_{wj}^t)^2}}) \implies \boldsymbol{\mu}_k^{ut} = \frac{\mathbf{r}_k^{ut}}{||\mathbf{r}_k^{ut}||_2}; \quad \boldsymbol{\sigma}_k^{ut} = \sigma^t \cdot exp(-\frac{1}{2}\mathbf{o}_k^{ut})$$

$$\mathbf{z}_k^{ut} \sim \mathcal{N}(\boldsymbol{\mu}_k^{ut}, [diag(\boldsymbol{\sigma}_k^{ut})]^2) \; \forall k = 1, 2, ..., J \tag{16}$$

Here, $g^t : \mathbb{R}^D \to \mathbb{R}^{2d}$ is function parameterized by neural network. $\sigma^t$'s value is around 0.1, following Ma et al. (2019b). Text factor $j$ representation of user $u$ is sampled from Gaussian distribution with estimated parameters. Assuming the independence between text factors of user $u$, we have $q(\mathbf{z}^{ut}|\mathbf{t}^u, \mathbf{A}^t) = \prod_{j=1}^J \mathcal{N}(\boldsymbol{\mu}_j^{ut}, [diag(\boldsymbol{\sigma}_j^{ut})]^2)$, which is called variational distribution, approximating intractable posterior distribution $p(\mathbf{z}^{ut}|\mathbf{t}^u, \mathbf{A}^t)$. $q(\mathbf{z}^{ut}|\mathbf{t}^u, \mathbf{A}^t)$ is matched with prior distribution $p(\mathbf{z}^{ut}) = \mathcal{N}(\mathbf{0}, (\sigma^t)^2 \mathbf{I})$ via Kullback-Leibler divergence ($D_{KL}^t$). As $p(\mathbf{z}^{ut})$ is factorized, $D_{KL}^t$ also imposes micro-disentanglement, i.e., disentangled between dimensions of representation sampled from $q(\mathbf{z}^{ut}|\mathbf{t}^u, \mathbf{A}^t)$. $D_{KL}^t(q(\mathbf{z}^{ut}|\mathbf{t}^u, \mathbf{A}^t)||p(\mathbf{z}^{ut}))$ will be plug in Equation 12 for optimization.

## A.2 Implementation

Training procedure of our proposed BANDVAE is presented in Algorithm 2.

For review purpose, we release code, data and related materials in the anonymized link https://drive.google.com/file/d/1GlhMB54tJHev0X7XeFw2fWbapy6tP274/view?usp=share_link

## A.3 Data Pre-processing

For Citeulike-a and Cell Phones datasets, we use the accompanying textual content, i.e., tile and abstract on Citeulike-a and item descriptions on Cell Phones. Additional step is performed on Cell Phones to retain users with at least 8 interactions and items with at least 5 interactions. For Movie-Lens, we follow Zhu & Chen (2022) to extract a subset of users from ML-10M version. We keep user ratings larger than 3 as interactions, following Ma et al. (2019b). We collect texts for items in Movielens from IMDB [4]. For all datasets, we remove stop words and only keep words with frequency higher than 3 and appearing in less than 60% of item texts. Following Zhu & Chen (2022), top $8k$ words with highest frequency are retained to construct vocabulary. Following Ma et al. (2019b), we adopt *strong generalization* setting to construct training, validation and test sets by randomly choosing 80% of users for training and 10% of users for each validation and test sets. For validation and test sets, 20% of a user interactions is kept as ground truth (test data). To keep the quality of datasets, we only retain items with at least 5 words in their textual content so that the textual content brings semantic information. All cold-start items, i.e., those do no appear in training set, are discarded since there is no parameters associating with them.

## A.4 Baseline description

We compare BANDVAE against a series of VAE-based recommendation models

---

[4]https://datasets.imdbws.com/

---

**Algorithm 2:** Training procedure of BANDVAE

---

**Input:**

- Rating vectors of $M$ users $\{\mathbf{y}^u\}_{u=1}^M$

- Text vectors of $M$ users $\{\mathbf{t}^u\}_{u=1}^M$

- Rating channel's parameters $\Theta^y$

    - item matrix in decoder $\mathbf{H} \in \mathbb{R}^{N \times d}$, context matrix $\mathbf{C}^y \in \mathbb{R}^{N \times D}$
    - prototype representations $\mathbf{m}^y \in \mathbb{R}^{K \times d}$
    - parameters neural network $g^y : \mathbb{R}^D \to \mathbb{R}^{2d}$

- Text channel's parameters $\Theta^t$

    - weight of decoder $\mathbf{E} \in \mathbb{R}^{W \times d}$, context matrix $\mathbf{C}^t \in \mathbb{R}^{W \times D}$
    - prototype representations $\mathbf{m}^t \in \mathbb{R}^{J \times d}$
    - parameters neural network $g^t : \mathbb{R}^D \to \mathbb{R}^{2d}$

- Parameter $\mathbf{W} \in \mathcal{R}^{2d \times 1}$ of fusion layer

- Hyper-parameters $\tau, \epsilon, \sigma^y, \sigma^t$

**Output:**

- Updated $\Theta^y$ and $\Theta^t$

1  **for** *batch user $\mathcal{B}$* **do**
2     **for** *user $u \in \mathcal{B}$* **do**
3         $\{\mathbf{z}_k^{uy}\}_{k=1}^K \leftarrow f_{rating}^{enc}(\mathbf{y}^u, \mathbf{m}^y, \mathbf{H}, \mathbf{C}^y, \tau, \sigma^y)$ // Rating encoder in Section 4.1
4         $\{\mathbf{z}_j^{ut}\}_{j=1}^J \leftarrow f_{text}^{enc}(\mathbf{t}^u, \mathbf{m}^t, \mathbf{E}, \mathbf{C}^t, \tau, \sigma^t)$ // Text encoder in Section 4.1
5         $\pi^u \leftarrow Sinkhorn\ algorithm(\{\mathbf{z}_k^{uy}\}_{k=1}^K, \{\mathbf{z}_j^{ut}\}_{j=1}^J, \epsilon)$ // Alignment coupling probability derivation in Algorithm 1
6         $\{\hat{\mathbf{z}}_k^{uy}\}_{k=1}^K \leftarrow Barycentric\ mapping(\pi^u, \{\mathbf{z}_j^{ut}\}_{j=1}^J)$ // Equation 7
7         $\{\hat{\mathbf{z}}_j^{ut}\}_{j=1}^J \leftarrow Barycentric\ mapping(\pi^u, \{\mathbf{z}_k^{uy}\}_{k=1}^K)$ // Equation 8
8         $\tilde{\mathbf{z}}^{uy} \leftarrow Fusion\ layer(\{\mathbf{z}_k^{uy}\}_{k=1}^K, \{\hat{\mathbf{z}}_k^{uy}\}_{k=1}^K)$ // Equation 9
9         $\tilde{\mathbf{z}}^{ut} \leftarrow Fusion\ layer(\{\mathbf{z}_j^{ut}\}_{j=1}^J, \{\hat{\mathbf{z}}_j^{ut}\}_{j=1}^J)$ // Equation 10
10        $\mathcal{L}_u^y \leftarrow Calculate\ rating\ channel\ loss$ // Equation 12
11        $\mathcal{L}_u^t \leftarrow Calculate\ text\ channel\ loss$ // Equation 14
12        Calculate regularization term $\mathcal{L}_u^{OT}$ // Equation 6
13     Calculate loss $\mathcal{L} = \frac{1}{||\mathcal{B}||}\sum_{u \in \mathcal{B}} \mathcal{L}_u^y + \lambda_t \cdot \mathcal{L}_u^t + \lambda_r \cdot \mathcal{L}_u^{OT}$
14     Update $\Theta^y, \Theta^t$ to minimize $\mathcal{L}$

Table 6: Effect of $\epsilon$ in Algorithm 1

| $\epsilon$ value | Citeulike-a | | Cell Phones | | MovieLens | |
|---|---|---|---|---|---|---|
| | R@10 | N@10 | R@10 | N@10 | R@10 | N@10 |
| 0.01 | 22.78 | 23.68 | 6.18 | 5.00 | 14.01 | 12.81 |
| 0.02 | 22.84 | 23.71 | 6.11 | 4.96 | 13.97 | 12.74 |
| 0.05 | 23.20 | 24.11 | 6.18 | 5.01 | 14.00 | 12.76 |
| 0.1 | 23.46 | 24.24 | 6.14 | 5.00 | 14.03 | 12.92 |
| 0.2 | **23.80** | **24.55** | 6.18 | 4.99 | 13.97 | 12.87 |
| 0.5 | 23.54 | 24.44 | **6.21** | **5.06** | 14.25 | 13.04 |
| 1 | 23.48 | 24.41 | 6.19 | 5.06 | **14.40** | **13.15** |
| 2 | 23.58 | 24.50 | 6.19 | 5.02 | 14.20 | 13.01 |
| 5 | 23.53 | 24.46 | 6.17 | 5.03 | 14.21 | 13.00 |

- **MacridVAE** Ma et al. (2019b) introduces macro- and micro-disentanglement of user presentation via multi-prototype representation and independence regularization.
- **RecVAE** Shenbin et al. (2020) proposes composite prior, rescaling regularization term and an alternative training approach to improve VAE-based recommendation model.
- **MDCVAE** Zhu & Chen (2022) regularizes decoder weights of user-oriented autoencoder by latent embeddings inferred from textual content.
- **TopicVAE** Guo et al. (2022) improves disentangling user representation by designing attention-based topic extraction from textual content, topic-guided contrastive loss and heuristic method to set value of regularization term.
- **ADDVAE** Tran & Lauw (2022) leverages two disentangled networks to model user's ratings and user associated texts then aligns disentangled factors from these two modalities using compositional de-attention and regularization.
- **SEM-MacridVAE** Wang et al. (2023a) exploits semantic knowledge from side information to improve disentangled recommendation model. We use tf-idf item-word matrix as side information for fair comparison.

We follow the strong generalization setting in MacridVAE Ma et al. (2019b), i.e., validation and test sets include users not appearing in training set. Thus, only VAE baselines are considered as they are capable of predicting interactions for new users.

## A.5 Hyper-parameter Tuning

Regarding BANDVAE, the default settings are $D = 300$ for MovieLens and Cell Phones and $D = 600$ for Citeulike-a; embdding size $d = 100$ for all datasets; dropout rate applied for $\mathbf{A}^y$ and $\mathbf{A}^t$ is 0.5; number of factors/prototypes/interests $K = 4$; $\beta = min(\beta_0, \frac{update}{T})$ where $\beta_0$ is 1 for rating channel and $\beta_0 = 0.2$ for text channel, $T$ is chosen from $\{1k, 5k, 10k, 20k\}$, and $update$ is the number parameter update; $\sigma_0^y$ and $\sigma_0^t$ are chosen from $\{0.05, 0.075, 0.1\}$; default choices of $\lambda_t$ is 0.1 for Citeulike-a, 0.5 for MovieLens and 0.5 for Cell Phones; $\lambda_r$ is 2 for Citeulike-a and $\lambda_r = 1$ for the rest; $\epsilon \in \{0.2, 0.5, 1\}$ in Sinkhorn algorithm. We train BANDVAE using Adam optimizer with learning rate 0.001 on NVIDIA RTX 2080 Ti GPU machine. Training stops after 30 epochs without improving performance on validation set.

## A.6 Effect of $\epsilon$ in Equation 4.

$\epsilon$ controls the sparsity of $\pi^u$, i.e., small $\epsilon$ results in highly skewed distribution in $\pi^u$ while large $\epsilon$ leads to roughly uniform distribution in $\pi^u$. Thus, $\epsilon$ controls the trade-off between *interpretability* and *recommendation*. That is, small $\epsilon$ generates a near one-to-one mapping between rating and text factors. In this case, we can explain one factor from ratings by the matched one from texts. Contrarily, large $\epsilon$ produces roughly one-to-many mapping between factors from two modalities, which causes difficulties when one would like to explain rating factor in terms of text. Table 6 reports the results. There are two key takeaways. First, there is a trade-off between recommendation

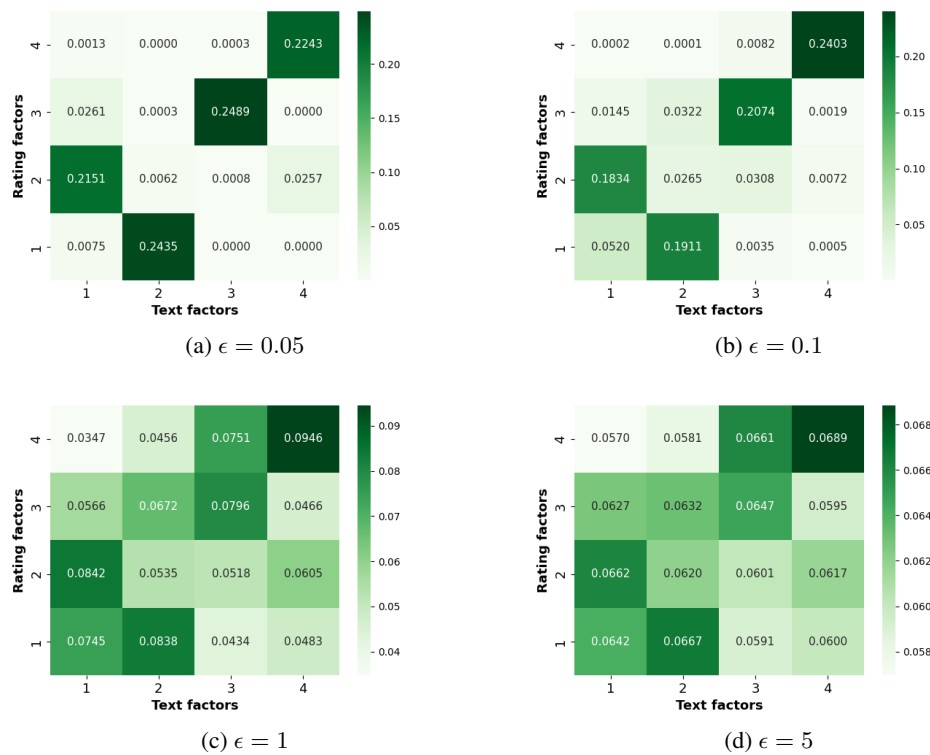

Figure 4: Alignment probability w.r.t $\epsilon$ of a user on Cell Phones dataset.

and interpretability. While recommendation task favors moderate $\epsilon$, i.e., larger than $0.2$, small $\epsilon$ which benefits interpretability does negatively affect recommendation performance. Second, $\epsilon$'s value is data-dependent and thus requiring careful analysis to achieve good performance.

**Effect of $\epsilon$ on alignment probabilities.** Theoretically, small $\epsilon$ results in sparse alignment distribution while large $\epsilon$ leads to roughly uniform alignment distribution of $\pi^u$ in Equation 5. Note that sparse alignment distribution is favorable to interpretability as it mimics approximately one-to-one mapping between rating and text factors. To verify this, we visualize the alignment probability produced by our model BANDVAE w.r.t various $\epsilon$ in Figure 4. Obviously, small $\epsilon$ results in staggered pattern in alignment probability while large $\epsilon$ leads to roughly uniform distribution. Thus, there is trade-off between interpretability and recommendation should be taken into account when aligning interest factors from ratings and texts.

### A.7 ANALYSIS ON THE NUMBER OF FACTORS

Thanks to the pair-wise alignment between interest factors, BANDVAE can deal with the case when the number of user interests between two modalities differ. As users might demonstrate different behaviors in different modalities, this capability makes BANDVAE more applicable. We report recommendation accuracy w.r.t. various number of rating factors $K$ and number of text factors $J$ in Figure 5. There are three data-dependent observations. First, on Citeulike-a, setting the number of rating and text factors both to 4 generally results in better recommendation accuracy. More than 6 interest factors lead to negative effect. Second, on Cell Phones, it requires at least $4$ factors to model user interests from ratings. While increasing the number of rating factors does benefit, adding more text factors generally does not help much. Adding more rating factors brings benefits to Recall at top 10 and NDCG at top 10 and top 50 yet causes negative effect on Recall at top 50. Third, on MovieLens, while adding more rating factors generally brings benefits, the number of text factors is around 3 or 4 to achieve highest accuracy. Exaggerated number of text factors negatively affects performance.

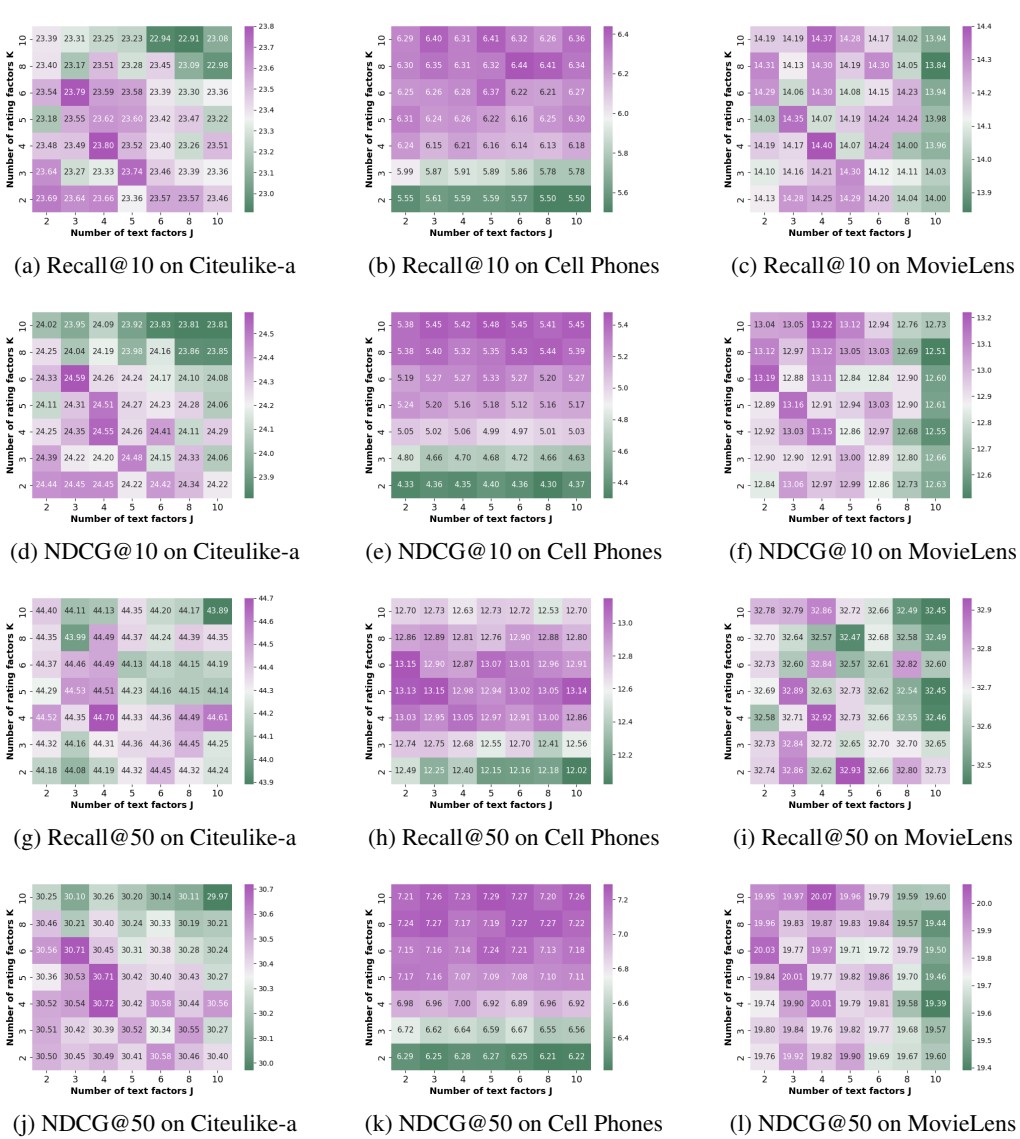

Figure 5: BANDVAE's performance w.r.t. $K$ and $J$.

Table 7: Effect of $\lambda_r$ on recommendation accuracy

| $\lambda_r$ value | Citeulike-a | | | | Amazon Cell Phones | | | | MovieLens | | | |
|---|---|---|---|---|---|---|---|---|---|---|---|---|
| | R@10 | R@50 | N@10 | N@50 | R@10 | R@50 | N@10 | N@50 | R@10 | R@50 | N@10 | N@50 |
| 0 | 23.31 | 44.38 | 24.28 | 30.51 | 6.10 | **13.05** | 4.95 | 6.93 | 14.19 | 32.63 | 12.97 | 19.82 |
| 0.1 | 23.42 | 44.38 | 24.32 | 30.53 | 6.20 | 12.89 | 5.03 | 6.93 | 14.18 | 32.59 | 12.98 | 19.82 |
| 0.2 | 23.48 | 44.44 | 24.39 | 30.61 | 6.20 | 12.93 | 5.03 | 6.95 | 14.15 | 32.60 | 13.00 | 19.83 |
| 0.5 | 23.54 | 44.53 | 24.34 | 30.61 | **6.24** | 12.99 | 5.02 | 6.93 | 14.31 | 32.82 | 13.07 | 19.93 |
| 1 | 23.51 | 44.47 | 24.41 | 30.62 | 6.21 | **13.05** | **5.06** | **7.00** | **14.40** | **32.92** | **13.15** | **20.01** |
| 2 | **23.80** | **44.70** | **24.55** | **30.72** | 6.11 | 12.98 | 5.00 | 6.94 | 14.21 | 32.74 | 12.92 | 19.81 |
| 5 | 23.52 | 44.66 | 24.39 | 30.68 | 6.04 | 13.00 | 4.94 | 6.93 | 14.14 | 32.91 | 12.96 | 19.91 |

Table 8: Effect of $\lambda_t$ on recommendation accuracy

| $\lambda_t$ value | Citeulike-a | | | | Cell Phones | | | | MovieLens | | | |
|---|---|---|---|---|---|---|---|---|---|---|---|---|
| | R@10 | R@50 | N@10 | N@50 | R@10 | R@50 | N@10 | N@50 | R@10 | R@50 | N@10 | N@50 |
| 0 | 23.52 | 44.68 | 24.36 | 30.64 | 6.19 | 12.69 | 4.98 | 6.83 | 14.07 | 32.59 | 12.82 | 19.69 |
| 0.1 | **23.80** | **44.70** | **24.55** | **30.72** | **6.27** | 12.90 | **5.07** | 6.95 | 14.16 | 32.78 | 12.98 | 19.86 |
| 0.2 | 23.46 | 44.50 | 24.27 | 30.53 | 6.16 | 12.84 | 4.95 | 6.86 | 14.11 | 32.74 | 12.89 | 19.77 |
| 0.5 | 23.24 | 44.47 | 24.20 | 30.51 | 6.22 | 12.81 | 5.02 | 6.90 | **14.40** | **32.92** | **13.15** | **20.01** |
| 1 | 23.10 | 44.41 | 24.04 | 30.40 | 6.21 | 12.94 | 5.03 | 6.95 | 14.19 | 32.58 | 12.96 | 19.78 |
| 2 | 22.95 | 44.31 | 23.93 | 30.31 | 6.21 | **13.05** | 5.06 | **7.00** | 13.99 | 32.76 | 12.79 | 19.77 |
| 5 | 22.97 | 44.42 | 23.97 | 30.38 | 6.07 | **13.05** | 5.01 | **7.00** | 13.70 | 32.70 | 12.34 | 19.44 |

## A.8 EFFECT OF $\lambda_r$ IN EQUATION 4.3

Table 7 presents the recommendation accuracy w.r.t. various values of $\lambda_r$, which controls the effect of regularization term for interest transfer between disentangled factors of two modalities. Generally, we observe that setting $\lambda_r$ to 1 or 2 results in highest accuracy on chosen datasets. These values boost model performance more than smaller ones do, showing that our proposed alignment probability guided regularization-based interest transfer is indeed beneficial to capture user interests from two modalities.

## A.9 EFFECT OF $\lambda_t$ IN EQUATION 4.3

Table 8 presents the influence of $\lambda_t$, which controls the effect of text reconstruction objective, on recommendation accuracy. The key observation is the effect of $\lambda_t$ is data-dependent, i.e., Citeulike-a prefers small $\lambda_t$ (0.1), MovieLens favors a slightly bigger one (0.5) and Cell Phones needs larger value (2) to achieve competitive accuracy. Thus, $\lambda_t$ should be chosen carefully to obtain the good performance.