# OpenReview forum: "Barycentric Alignment of Mutually Disentangled Modalities"
_ICLR.cc/2024/Conference — Submitted to ICLR 2024_

### Official Review · Reviewer_ZCpy · 2023-10-31

**Soundness:** 3 good
**Presentation:** 1 poor
**Contribution:** 3 good
**Rating:** 5
**Confidence:** 4

**Summary:**

The paper addresses the challenge of uncovering explanatory factors of user preferences by aligning behavioral data with side information from textual descriptions modality. The work tackles the complex problem of aligning user interest factors from unsupervised modalities. It introduces a novel approach called BANDVAE, treating discovered interest factors as discrete probability masses and formulating their alignment as an optimal transport problem. The alignment probability not only serves as prior information but also is used to fuse interest factors through a barycentric strategy, thus transferring knowledge about user preferences between the mutually disentangled modalities.

The text is well-structured and easy to read. The theory behind the proposed approach is presented clearly and coherently. The overall solution looks interesting and novel. Authors provide a comprehensive experimental verification on several benchmark datasets. The research has the potential to advance the understanding of user preference factors and their transfer between related modalities. However, I have several issues with the conceptualization of the problem solved by the authors and several related questions, which are provided below.

**Update**. The responses from the authors provide additional valuable context. However, the issue with obscuring a standard hybrid recommender system behind the multi-modality concept remains unresolved. The claim on the superiority of hybrid recommender systems over the standard collaborative filtering models does not have sufficient evidence, the authors overstate the importance of side features, which leads to incomplete experiments and testing scenarios (e.g., cold start). While I do see novelty and originality in the proposed approach, I'm lowering my vote for this paper after carefully reading the responses. The work should be properly positioned in the landscape of hybrid methods and provide better experimental evidence of the superiority over both hybrid methods and strong non-hybrid collaborative filtering approaches (for example, EASEr model, a linear shallow autoencoder by Harald Steck).

**Strengths:**

- a novel alignment strategy to extract and fuse knowledge about various aspects of user preferences with barycentric strategy
- solid mathematical foundation for the solution
- rich experimental base that provides a detailed look at the performance of the proposed approach

**Weaknesses:**

- the work implicitly relies on the assumption of the relatedness between behavioral data and side information, which in general is not supported by prior empirical evidence
- the work is focused on a specific type of side information - textual description, which limits its generality
- in important testing scenario related to cold-start settings is missing

**Questions:**

### **Main concerns**
In many places in the text, there's an implicit assumption that modalities are mutually related, e.g., "transfer knowledge between interest factors from mutually related modalities". This assumption may not be true in practice, i.e., modalities may be "non-alignable". It is especially true for side information, i.e., static descriptive features and attributes of users and items. The authors state that "side information could complement those mined from user consumption behaviors". So, the work implicitly assumes that ratings are not expressive enough. The verification of that requires very rigorous study. In the general case, focusing on side features mostly leads to the so-called overspecialization, meaning that side features are not descriptive enough to provide any additional knowledge on top of what's hidden in the behavioral data and can only limit the expressivity of models, not improve it. See, for example, a classical study on the "even few ratings are more valuable than metadata" effect by [Pilászy and Tikk 2009].  The exception is the cold-start scenario, where no preferences are provided and only side features are present, which apparently requires a mapping between collaborative and side information spaces. Application of hybrid models (which the proposed approach also belongs to) is logical in this scenario. Unfortunately, the authors do not provide any experimental results related to that scenario and only test their approach against the standard collaborative filtering task, which greatly limits the understanding of the utility of their approach.

I have some concerns about the interpretation part of the work as well. A common observation that the more interpretable the model is, the lower its quality. This observation is also confirmed in the current work, where the authors say that smaller $\epsilon$ that favor interpretability leads to worse recommendations. This is also in line with the aforementioned overspecialization remark. On the other hand, the proposed architecture still puts a lot of focus on interpretability. For example, according to the authors, "turning off effect of $\mathcal{L}^t_u$ hurts recommendation accuracy, showing its benefit." This claim is not directly supported by empirical evidence. The results provided to support this claim show no statistical significance. The visible improvement may be simply due to overfitting to specific factor (like popular words) or some other training/data-splitting artifacts that lead occasionally higher values because the model is sensitive to those factors. I also miss the confidence intervals for these data.

In addition to that, no comparison is made with simpler modes, i.e. Factorization Machines (FM) by [Rendle 2010]. The adherence between items and features, which is used by the authors to highlight interpretability, can be analyzed via inner products of the item and feature vectors learned by FM. So, the interpretability aspect cannot be considered novel here, although, the approach to obtain representations to be analyzed is different. Overall, the utility of the interpretation part of the architecture (text decoder) is not convincing from the empirical perspective and underscoring its importance generally contradicts common observations. It seems that the model learns just fine without it. Maybe there would be a difference in the cold-start scenario, but no empirical results are provided for this case.

The text also suffers from excessive generalization of the results and conclusions, starting even from the title of the paper. The title is too general, and the promise of the title doesn't correspond to the provided results. Only textual factors are considered in the work, which is not the general case of arbitrary modalities. No other features were considered. User features are also out of context. I'd suggest reformulating the title and changing the text in the introduction and conclusion accordingly to better reflect the limitations of the work and its applicability.

Moreover, there's already the term for the type of models that mix behavioral and side information - "hybrid models" (such as FM,  HybridSVD [Frolov and Oseledets 2019], or "hybrid" EASEr [Jeunen at el. 2020]). A clear connection must be made to this family of models, as the term "modalities" is not used there while the same general task is being solved.

### **Other issues**
- Figure 1 lacks notation, adding and denoting the different weight matrices on the provided scheme would help to better understand the approach.

- $(r^{uy}_k , o^{uy}_k )$ -  the notation is not clear, is it a concatenation?

- Movielens dataset is described as: 15,000 users; 7,892 items; 1,005,820 interactions; 8,000 words.  There are multiple Movielens datasets, which one exactly is used? It doesn't match the dimensions of the standard Movielens-1M dataset.  Also, having a dedicated table with dataset statistics would improve the methodological part of the work.

- Only one training epoch is analyzed in terms of computational efficiency. This information is not sufficient. How many epochs are required to train an optimal model in each case? More substantial analysis is required. A comparison with simpler models that do not use side features is also required. The marginal gains of using side features may not be worth the increased computational demands.

### **References**
Pilászy, I. and Tikk, D., 2009. Recommending new movies: even a few ratings are more valuable than metadata. In _Proceedings of the third ACM conference on Recommender systems_ (pp. 93-100).

Rendle, S., 2010, December. Factorization machines. In _2010 IEEE International conference on data mining_ (pp. 995-1000). IEEE.

Frolov, E. and Oseledets, I., 2019, September. HybridSVD: when collaborative information is not enough. In _Proceedings of the 13th ACM conference on recommender systems_ (pp. 331-339).

Jeunen, O., Van Balen, J. and Goethals, B., 2020, September. Closed-form models for collaborative filtering with side-information. In _Proceedings of the 14th ACM Conference on Recommender Systems_ (pp. 651-656).

---

> ### Author Response · Authors · 2023-11-15
> **Responses to Reviewer ZCpy (Part 1/3)**
>
> Dear Reviewer **ZCpy**,
>
> We would like to thank you for your time reviewing our paper. We really appreciate your detailed and constructive comments. Your concerns are addressed as follows
>
> **Q: the work implicitly relies on the assumption of the relatedness between behavioral data and side information, which in general is not supported by prior empirical evidence.**
>
> **A:** Recent studies [2, 3, 4] have shown that incorporating side information such as textual and/or visual content is beneficial to recommendation models, e.g., improving recommendation accuracy. As collected rating data is often sparse in nature, which could prevent learning expressive user/item representations, side information provides additional semantic information of users/items, which contributes to learn higher quality of user and item embeddings from both rating data and side information.
>
> **Q: Modalities may be non-alignable. … The work implicitly assumes that ratings are not expressive enough...
> Focusing on side features mostly leads to the so-called overspecialization, meaning that side features are not descriptive enough to provide any additional knowledge on top of what's hidden in the behavioral data and can only limit the expressivity of models, not improve it**
>
> **A:** We would like to make two clarifications here. *First*, recent studies [2, 3, 4] have demonstrated that side information helps alleviate rating data sparsity problem to achieve higher recommendation accuracy.
>
> *Second*, in our model, textual content does not completely replace ratings to make recommendations. Rather, it acts as an accompaniment. We fuse user interest factors learned from textual content with those from ratings to make rating interest factors more expressive. That it achieves higher recommendation accuracy than baselines, which evidences that the modalities are aligned.
>
> **Q: Discussion on a classical study on the *even few ratings are more valuable than metadata* effect by [1]**
>
> **A:** In [1], the authors concluded that *“... even 10 ratings of a new movie are more valuable than the best solely metadata-based representation. ... this is due to the large gap between the movie descriptions and the movies themselves: people rate movies, not their descriptions”*
>
> The experiment leading to this conclusion is that the authors of [1] compare the proposed model using ratings against another model that only uses metadata to make content-based recommendation. Again, we would like to emphasize that our model does not fully rely on textual content to make recommendations. We still rely on rating data to make recommendations, which shares the same intention with [1]. The novelty is we innovatively fuse user interest factors learned from texts with those learned from ratings in disentangled fashion using alignment method (Section 4.2.1) and mapping method (Section 4.2.2) to obtain more expressive user representations.
>
> **Q: On the argument with respect to cold-start scenario**
>
> **A:** We agree that side information, e.g., visual or textual content, can be used to resolve cold-start problem in recommender systems. However, the key research question in this paper is how to fuse disentangled user interest factors from texts with those learned from ratings to achieve a more accurate recommendation model. Recent studies on multi-modal recommender systems also focus on how to leverage side information to obtain more expressive representations, leading to higher recommendation accuracy.
>
> As our focus is not resolving the cold-start problem, we do not dive into cold-start scenario to keep this paper focused on disentangled representation learning with textual content.
>
> [1] Pilászy, I. and Tikk, D., 2009. Recommending new movies: even a few ratings are more valuable than metadata. In Proceedings of the third ACM conference on Recommender systems (pp. 93-100).
>
> [2] Truong et al. Multi-Modal Recommender Systems: Hands-On Exploration. ACM RecSys 2021.
>
> [3] Wu et al. A Survey on Accuracy-Oriented Neural Recommendation: From Collaborative Filtering to Information-Rich Recommendation. TKDE 2023.
>
> [4] Zhou et al. A Comprehensive Survey on Multimodal Recommender Systems: Taxonomy, Evaluation, and Future Directions. https://arxiv.org/pdf/2302.04473.pdf

---

> ### Author Response · Authors · 2023-11-15
> **Responses to Reviewer ZCpy (Part 2/3)**
>
> **Q: Smaller $\epsilon$ that favors interpretability leads to worse recommendations. This is in line with the overspecialization remark**
>
> **A:** We would like to clarify that the trade-off between interpretability and recommendation in our model is controlled by $\epsilon$ in Equation 4, which controls the sparsity of alignment probabilities in Algorithm 1.
>
> When the alignment probabilities are close to uniform distribution (large $\epsilon$), it is difficult to find a one-to-one mapping between rating and text factors of a single user, thus decreasing the interpretability because we do not know which rating interest factors align with which text interest factors. When $\epsilon$ is small, the alignment probabilities are highly skewed, we can easily find a one-to-one mapping between rating interest factors and text interest factors. Therefore, we can understand the meaning of a rating interest factor via the aligned text factor by visualizing the output of text decoder corresponding to the aligned text factor.
>
> We illustrate this concept in Figure 4 in supplementary materials.
>
> **Q: The proposed architecture still puts a lot of focus on interpretability**
>
> **A:** For clarification, we focus on disentangled representation learning to improve recommendation accuracy as this direction has been shown to be effective [1].  The interpretability is embedded inside disentangled representations by nature [2] as it discovers factors of variation underlying data. We transfer this concept from disentangled representation learning to recommender systems to discover user interest factors from ratings and texts. Thus, interpretability is naturally incorporated. Interpretability is not a new feature, rather we analyze it to further understand the proposed model, i.e., how the proposed aligning and fusing method relates to interpretability.
>
> **Q: Effect of $\mathcal{L}^t_u$ due to overfitting to specific factor (like popular words), or training/data-splitting artifacts that lead occasionally higher values**
>
> **A:** To clarify the understanding, turning off the effect of $\mathcal{L}^t_u$ means the model does not have the supervision signals from textual content. Then the model is free to embed anything into text factors. Table 5 demonstrates that when including $\mathcal{L}^t_u$, text factors are trained to capture interest factors from texts, which is more meaningful than freely embedding anything into them, leading to better recommendation accuracy. Regarding statistical significance, in Table 5, the numbers of Citeulike-a R@10 and MovieLens R@10 and N@10 are statistically significant. Table 5 brings two key takeaways. For one, textual signals are helpful to recommendation accuracy to a certain extent. For another, the improvement of BandVAE not only comes from simply reconstructing textual content but it is also (mainly) from the proposed aligning and fusing modules, which are the key contributions in this paper.
>
> **Q: no comparison is made with simpler modes, i.e. Factorization Machines (FM) by [Rendle 2010]**
>
> **A:** We did not compare BandVAE against FM as one of the baselines MDCVAE has outperformed FM. As the main target is to improve recommendation accuracy via disentangled representation learning, we involve recently developed disentangled recommendation models (and these models already demonstrated state-of-the-art results), including both those using textual content and those that do not, to verify the advantages of our model.
>
> **Q: The interpretability aspect cannot be considered novel here, although, the approach to obtain representations to be analyzed is different**
>
> **A:** As stated in the contributions, the key novelty in this paper is how to align and fuse disentangled user interest factors uncovered from ratings and texts.  The interpretability is brought by disentangled representation learning by nature [2]. We analyze interpretability to further understand the inner working of the proposed model.
>
> **Q: The title is too general. Only textual factors are considered in the work, which is not the general case of arbitrary modalities**
>
> **A:** We consider two modalities ratings and texts. Ratings are the key modality in multi-modal recommender systems [3]. A more specific rephrasing of the title could be *Barycentric Alignment of Mutually Disentangled Rating and Text Modalities*.
>
> Other modalities can be incorporated in our model. The alignment method can work for two mutually disentangled modalities. For example, one can replace the text channel with an image channel of which the inputs are image pixels. We consider textual content as it is available in chosen datasets while images are not available in Citeulike-a or MovieLens.
>
> [1] Ma et al. Learning Disentangled Representations for Recommendation. NeurIPS 2019.
>
> [2] Bengio et al. Representation learning: A review and new perspectives. TPAMI 2013.
>
> [3] Truong et al. Multi-Modal Recommender Systems: Hands-On Exploration. ACM RecSys 2021.

---

> ### Author Response · Authors · 2023-11-15
> **Responses to Reviewer ZCpy (Part 3/3)**
>
> **Q: A clear connection must be made to family of hybrid models (FM, hybridSVD, hybrid EASEr)**
>
> **A:** Our proposed model shares the intention with hybrid models (FM, hybridSVD, hybrid EASEr), combining techniques from content-based filtering and collaborative filtering to achieve higher recommendation accuracy. We would consider adding this discussion in the final version.
>
> **Q:** **Figure 1 lacks notations**
>
> **A:** We already provided key notations in Figure 1. Due to limited space, the detailed descriptions of weight matrices are presented in preliminary sections to ease the reading.
>
> **Q: The notation $(r^{uy}_k, o^{uy}_k)$ is not clear, is it a concatenation?**
>
> **A:** This notation means the output includes two terms. We follow MacridVAE [1] for consistent notations.
>
> **Q: MovieLens dataset statistics**
>
> **A:** As described in supplementary, the MovieLens dataset is a subset of MovieLens-10M. We only include items with textual content, and pre-process data following MDVAE [2].
>
> **Q: On the evaluation of computational efficiency**
>
> **A:** For clarification, to verify time efficiency we report the training time to run a single epoch in our GPU server in the main text. The reported time is averaged over multiple runs. The main observation is that the proposed BandVAE only requires slightly higher training and inference time than the closest baseline ADDVAE.
>
> The table below shows the number of training epochs to achieve the reported results. We run each model 5 times with different random seeds on each dataset then report the averaged number of required epochs. For contrasting, we compare BandVAE against ADDVAE which uses textual content and MacridVAE which does not use textual content. There are two key takeaways. *First*, models using textual content does not always require more training epoch than models not using text content, which is shown by contrasting MacridVAE and ADDVAE. *Second*, BandVAE only requires slightly more training epochs than both MacridVAE and ADDVAE yet achieves higher recommendation accuracy, demonstrating the effectiveness and efficiency of BandVAE.
>
> |           | Citeulike-a | MovieLens | Cell Phones |
> |:---------:|:-----------:|:---------:|:-----------:|
> |  BandVAE  |     58      |    45     |     34      |
> |  ADDVAE   |     50      |    38     |     33      |
> | MacridVAE |     46      |    40     |     36      |
>
> [1] Ma et al. Learning Disentangled Representations for Recommendations. NeurIPS 2019.
>
> [2] Zhu et al. Mutually-Regularized Dual Collaborative Variational Auto-encoder for Recommendation Systems. The Web Conference (WWW) 2022.

---

> ### Author Response · Authors · 2023-11-20
> **Looking forward to your feedback**
>
> Dear Reviewer **ZCpy**,
>
> This is a friendly reminder that whether our responses have addressed your concerns and questions.
>
> We are happy to discuss if you have further questions.
>
> Best Regards,
>
> Authors.

---

### Official Review · Reviewer_Mi6r · 2023-11-06

**Soundness:** 3 good
**Presentation:** 3 good
**Contribution:** 3 good
**Rating:** 8
**Confidence:** 3

**Summary:**

The paper proposes a method to disentangle and align interest factors across modalities for recommendation. The model is named BandVAE, short for Barycentric Alignment of Mutually Disentangled Modalities with Variational AutoEncoder. First, the specific interest factors from two modalities (i.e., ratings and textual descriptions of items) are uncovered. Second, as such factors are regarded as supporting points of two discrete measures, the authors perform their alignment by means of optimal transportation (OT). Third, two approaches are proposed for the knowledge transfer problem, namely: 1) a regulariser term is added to the loss function to guide alignment, and 2) barycentric mapping is adopted to map ratings to the textual content of items and vice versa, while the two signals are reconstructed through a decoder to allow the knowledge transfer across the two modalities. The proposed approach is tested on three popular recommendation datasets against seven VAE-like recommendation models from the literature, demonstrating the efficacy of the BandVAE solution. Finally, several ablation studies and hyper-parameter value analyses, along with an interpretability assessment of the model, further motivate the goodness of BandVAE.

**Strengths:**

+ Using optimal transportation in the case of mutual alignment of modalities for recommendation is relatively new in the field.
+ The formalization of the method is adequately sound.
+ The authors release the code at review time in the supplementary materials.
+ The experimental setting is extensive with several evaluation dimensions.
+ The interpretability analysis is quite helpful to further justify the efficacy of the approach.

**Weaknesses:**

- Ratings are not usually regarded as one modality in recommendation.
- With reference to Table 1, it seems that the improvement over the tested baselines is not sufficiently large to justify the efficacy of the proposed approach; the authors may provide additional results (if any) to effectively test BandVAE’s performance against the other baselines.

**After rebuttal.** All concerns have been addressed by the authors in the rebuttal.

**Questions:**

* Why did the authors decide to adopt ratings as one of the modalities involved in recommendation? A common approach in recommendation leveraging multimodalities is to exploit visual and textual features of product images, which might represent an interesting direction to follow.
* Can the authors provide justifications for the not-so-high improvement with respect to the tested baselines in Table 1?

**After rebuttal.** All concerns have been addressed by the authors in the rebuttal.

---

> ### Author Response · Authors · 2023-11-15
> **Responses to Reviewer Mi6r**
>
> Dear Reviewer **Mi6r**,
>
> We appreciate your constructive comments and your time reviewing our paper. We address your concerns as follows:
>
> **W1:** **Ratings are not usually regarded as one modality in recommendation**
>
> **A1:** Ratings are considered as the key modality in multi-modal recommender systems [1]. Textual content, visual content, etc., are considered as other modalities to alleviate data sparsity problem, which helps recommendation models learn more expressive embeddings. As rating data is often highly sparse in nature, learning user/item representations from other data-modalities, e.g., texts or images, then fusing them with those learned from ratings has been shown to alleviate the data sparsity problem and obtain higher quality of user and item representations [1, 2].
>
> [1] Truong et al. Multi-Modal Recommender Systems: Hands-On Exploration. ACM RecSys 2021.
>
> [2] Zhou et al. A Comprehensive Survey on Multimodal Recommender Systems: Taxonomy, Evaluation, and Future Directions. https://arxiv.org/pdf/2302.04473.pdf
>
>
> **Q1:** **Why did the authors decide to adopt ratings as one of the modalities involved in recommendation?**
>
> **A1:**  Ratings serve as key information to make recommendations. Existing recommendation models adopt ratings to learn user and item embeddings then make recommendations based on the learned embeddings. Our paper follows this typical paradigm in recommender systems research.
>
> Textual content is widely available in popular benchmark datasets such as Citeulike, MovieLens and Amazon datasets (visual content is not available in Citeulike and MovieLens), so we adopt textual content to complement rating data, i.e., fusing user representations learned from texts with those learned from ratings to derive more expressive user representations.
>
> A future work could extend our proposed method to incorporate visual content by finding its projection onto rating space via barycentric mapping strategy then combining the projected embeddings of visual content with those from texts and ratings (as described in Section 4.2.2. Mapping and Fusing) to derive user embeddings, which could be more expressive representations and has the potential to improve recommendation performance.
>
> **Q2:** **It seems that the improvement over the tested baselines is not sufficiently large? Can the authors provide justifications for the not-so-high improvement with respect to the tested baselines in Table 1?**
>
> **A2:** The results from Table 1, which are averaged over 10 runs, show that the proposed BandVAE achieves statistically significantly higher recommendation accuracy than the closest performing model on Citeulike-a and Cell Phones datasets, demonstrating the efficacy of BandVAE.
>
> On MovieLens dataset, RecVAE achieves the second highest accuracy (after our method BandVAE). That RecVAE is based on the complex multi-layered encoder (which uses roughly 1.2M parameters (around 18% of RecVAE’s parameters) more than encoder of BandVAE) and the composite prior layer. On other datasets, RecVAE is much less competitive.

---

> > ### Comment · Reviewer_Mi6r · 2023-11-20
> >
> > Dear Authors,
> >
> > thank you for your rebuttal. In the following, I'll provide an answer for each of the points addressed in the rebuttal.
> >
> > **W1/Q1.** Thank you for the references you provided. Indeed, some of the existing literature indicates that ratings may be considered as another modality. My concern was mainly related to the fact that the proposed approach (BandVAE) might have been trained in an _explicit_ feedback scenario (i.e., having explicit ratings in the dataset), while the other baselines have been trained in an _implicit_ feedback scenario. If this is the case, the training of BandVAE could have been boosted with respect of the other baselines (trained with only 0/1 users' feedback). If this is not the case (i.e., all models have been trained in an _explicit_ feedback scenario) my concern may be taken off.
> >
> > **W2/Q2.** Thank you for your clarification regarding the 10 rounds of training. Could you please also explain which statistical significance method you used, and with which thresholds?

---

> ### Author Response · Authors · 2023-11-20
> **Have we addressed your concerns?**
>
> Dear Reviewer **Mi6r**,
>
> We just want to post a friendly reminder that whether our responses have addressed your concerns.
>
> If you have further questions, please feel free to post. We are looking forward to hearing from you.
>
> Best Regards,
>
> Authors

---

> ### Author Response · Authors · 2023-11-20
>
> Dear Reviewer **Mi6r**,
>
> Thank you for your responses. We address your concerns as following
>
> **W1/Q1:** For fair comparison, all models, including the proposed BandVAE and baselines, were trained using *implicit feedback* data in our experiments.
>
> Although the proposed BandVAE and baselines can work directly on *explicit feedback* data, e.g., explicit ratings from 1 to 5, we choose *implicit feedback* data as the literature has suggested that a recommendation model with good rating prediction performance, measured by RMSE/MAE on observed ratings, might not perform well on recommendation task [1] in many cases. Therefore, one possible direction is to work on *implicit feedback* data and evaluate top-N recommendation performance using accuracy metrics such as Recall, which is what we did in this paper.
>
> [1] Cremonesi et al. Performance of recommender algorithms on top-n recommendation tasks. ACM RecSys'10.
>
>
> **W2/Q2** We measure statistical significance using a paired t-test with p-value < 0.05, following ADDVAE's setting. There exist cases when numbers are statistically significant with smaller p-value, e.g., p-value < 0.01.

---

> > ### Comment · Reviewer_Mi6r · 2023-11-20
> >
> > In the following, my further answers to your answers.
> >
> > ---
> >
> > **W1/Q1.** My concern still remains because, even though all the models were trained in an _implicit_ feedback setting (for fair comparison), BandVAE uses _explicit_ feedback as one of the multimodal features. Would it be not fair with respect to the other baselines, which do not have access to such piece of information in any phase of the training and evaluation?
> >
> > I'll try to explain this even better. Using multimodal features as side information (e.g., visual, textual, audio features) in my view (and according to the literature) is fine with respect to the other non-multimodal baselines, since one of the strengths of multimodal-aware models is to enhance the items' representation in this manner. However, using _explicit_ feedback as side information would not be fair with respect to the other baselines trained and tested in an _implicit_ setting. That's the reason why I asked whether it is fine to consider rating as one modality in the first place. I know the recent literature states so, but (personally) I'm not entirely convinced (mostly for the reasons I explained above).
> >
> > ---
> >
> > **W2/Q2.** Thank you for your answer. I'm sufficiently satisfied.

---

> ### Author Response · Authors · 2023-11-20
>
> Dear Reviewer **Mi6r**,
>
> Thank you for your explanation.
>
> **W1/Q1.** We would like to clarify that *BandVAE does not use any explicit feedback data, and explicit feedback data is **not** a side information (i.e., a multimodal feature) of BandVAE*.
>
> As described in Section 3 (PRELIMINARIES AND NOTATIONS), the inputs of BandVAE only include $y^u$, which is the binary rating vector (implicit feedback) of user $u$, and $t^u$, which is the textual content associated with user $u$. BandVAE only uses these information to make recommendations, which is fair when comparing with baselines.
>
> We disentangle user preference factors from $y^u$ and $t^u$, then aligning them using the proposed optimal transport-based approach as described in Section 4 to derive more informative user representations, leading to better recommendation accuracy.
>
> Binary rating data (implicit feedback) is also considered the key modality to make recommendations [1]. We, thus, also refer binary rating vector of users (implicit feedback) as one modality in BandVAE together with text modality.
>
> [1] Truong et al. Multi-Modal Recommender Systems: Hands-On Exploration. ACM RecSys 2021.

---

> > ### Comment · Reviewer_Mi6r · 2023-11-20
> >
> > Thank you for this further clarification. Now all my doubts have been addressed.
> >
> > I'll raise the rating accordingly.

---

> > > ### Author Response · Authors · 2023-11-21
> > >
> > > Dear Reviewer **Mi6r**,
> > >
> > > We are encouraged that your concerns have been addressed. We appreciate your updated evaluation score.
> > >
> > > Best Regards,
> > >
> > > Authors.

---

### Official Review · Reviewer_esB1 · 2023-11-06

**Soundness:** 2 fair
**Presentation:** 3 good
**Contribution:** 3 good
**Rating:** 6
**Confidence:** 3

**Summary:**

In this paper, the authors study the problem of aligning factors from disentangled modalities and ideate a novel method which aligns disentangled factors via a transfer knowledge method through guided regularization and barycentric mapping.

**Strengths:**

One of the strengths of this paper is that it addresses a very challenging problem. User signals/rating maps are very sparse and it often helps to mine other data modalities and then relate the two to get some relevant recommendations out of it.

**Weaknesses:**

Lacking some real-world applications of this approach.

**Questions:**

None

---

> ### Author Response · Authors · 2023-11-15
> **Responses to Reviewer esB1**
>
> Dear Reviewer **esB1**,
>
> We would like to thank you for spending time reviewing our paper. We address your concern as following
>
> **Q:** **Lacking some real-world applications of this approach**
>
> **A:** The proposed approach in this paper is tested on the recommender system, which is a real-world research problem. Not only is it applicable for rating and text modalities but the proposed method also works for alignment problems between disentangled factors from other mutually disentangled data modalities, e.g., ratings and images, images and texts.

---

> ### Author Response · Authors · 2023-11-20
> **Looking forward to your feedback**
>
> Dear Reviewer **esB1**,
>
> May we know that whether we have addressed your concern?
>
> If you need further clarification, we are happy to discuss.
>
> Best Regards,
>
> Authors.

---

### Meta-Review · Area_Chair_m5B1 · 2023-12-28

**Metareview:**

The authors proposed a method for: (1) (mutually) disentangling interest factors between modalities (in this case, ratings and text review content) for interpretability improvements and mitigating rating sparsity and (2) fusing the disentangled factors to improve recommendation performance (BandVAE). The key technical contribution is the use of an optimal transport formulation over the discrete latent interest factor distribution for fusing modalities to improve recommendations. Based on this formulation, the propose two approaches for the knowledge transfer problem: (1) guiding alignment with a regularizer term in the loss function and (2) a barycentric mapping between modalities with reconstruction during decoding. BandVAE is evaluated on three widely-used recommendation datasets (CiteULike, Amazon Cell Phones, MovieLens) and contrasted with several VAE-based RecSys methods, demonstrating consistent improvements against these (relatively strong) baselines. Finally, multiple secondary studies are performed including an ablation study, sensitivity analysis, efficiency study, and interpretability assessment to highlight specific properties of the BandVAE.

Consensus strengths identified by reviewers regarding this submission include:
- Optimal transport based methods for distributional alignment have gained interest in multiple settings. The authors apply it to RecSys, where is seems an ideal tool for mitigating rating sparsity while maintaining interpretability. Translating existing results from other works was non-trivial and the described formulation is well-principled (and emits nice theoretical properties by default).
- The empirical evaluation is convincing and extensive. Additionally, the discussion of the experiments highlights interesting points based on the author observations. Additionally, the interpretability study is illuminating.
- The paper is well-structured and the presentation of the proposed methods is mathematically rigorous.
- Reviewer concerns were generally well-addressed during the author rebuttal and discussion period (even if one reviewer didn't totally agree).

Conversely, consensus limitations included:
- While this is presented as a general method, the specific problem studied is ratings and review text. In my opinion, the idea of a plentifully available, sparse modality (e.g., ratings) and a less plentiful, dense modality (e.g., review text) does add some generality, but will require additional work to translate to such settings. This perspective does make the title sound a bit as over-claiming contributions.
- One reviewer raised the issue of evaluating in a cold-start scenario, which while not a focus of the paper, is also aligned with addressing rating sparsity with a second modality -- so it shouldn't be outright dismissed as it is of interest and at least discussed. However, it is reasonable to consider as future work and would make for a stronger paper as the proposed method should apply in transfer settings.
- In my opinion, the actual text of the writing needs further editing as there are many awkward sentence structures. I highly recommend a detailed editing in this regard.

Overall, the key contribution is using an optimal transport formulation to align rating and text modalities in RecSys. The paper is well-principled, the technical presentation is strong, and the empirical results are positive and well-discussed. While the paper is borderline, I do believe that it is useful for RecSys and the findings may be applicable to using optimal transport methods in other (non-RecSys) settings. Thus, I think it is applicable and the community would benefit from it being accepted, but also think it would benefit from another round of edits to address concerns.

**Justification For Why Not Higher Score:**

The work is focused on ratings and review text in RecSys settings. While I think the findings may translate back to other settings where optimal transport methods over latent discrete distributions, it isn't part of this work. Secondly, the focus of this paper was on a specific area of RecSys and didn't address some valid variants discussed by the reviewers.

**Justification For Why Not Lower Score:**

N/A

---

### Decision · Program_Chairs · 2024-01-16

Reject